# ACRF: Compressing Explicit Neural Radiance Fields via Attribute Compression

**Guangchi Fang**[1]**, Qingyong Hu**[2]**, Longguang Wang**[3]**, Yulan Guo**[4]
[1]Sun Yat-sen University, [2]University of Oxford, [3]Aviation University of Air Force,
[4]National University of Defense Technology

## Abstract

In this work, we study the problem of explicit NeRF compression. Through analyzing recent explicit NeRF models, we reformulate the task of explicit NeRF compression as 3D data compression. We further introduce our NeRF compression framework, Attributed Compression of Radiance Field (ACRF), which focuses on the compression of the explicit neural 3D representation. The neural 3D structure is pruned and converted to points with features, which are further encoded using importance-guided feature encoding. Furthermore, we employ an importance-prioritized entropy model to estimate the probability distribution of transform coefficients, which are then entropy coded with an arithmetic coder using the predicted distribution. Within this framework, we present two models, ACRF and ACRF-F, to strike a balance between compression performance and encoding time budget. Our experiments, which include both synthetic and real-world datasets such as Synthetic-NeRF and Tanks&Temples, demonstrate the superior performance of our proposed algorithm.

## 1 Introduction

In recent years, neural radiance fields (NeRFs) (Mildenhall et al., 2020) have emerged as a popular research topic due to its wide application and expressive performance in novel view synthesis, 3D reconstruction, and 3D perception. To improve the performance of NeRFs in terms of rendering quality and training/inference efficiency, recent research has explored the use of explicit 3D representations such as voxel grids (Sun et al., 2022; Chen et al., 2022a; Yu et al., 2022; 2021), point clouds (Xu et al., 2022; Ost et al., 2022), and meshes (Chen et al., 2023). These representations provide a stronger capacity for NeRFs to learn and represent 3D scenes by explicitly encoding the geometry and topology of the scene into a structured data format. However, the use of explicit 3D representations comes with a significant drawback, which is the substantial increase in model size, leading to challenges in transmitting and storing NeRF models. In this regard, we study the critical problem of explicit NeRF compression, which aims to reduce the size of a NeRF model while maintaining its rendering quality and other performance metrics.

Several previous works (Deng & Tartaglione, 2023; Li et al., 2023; Rho et al., 2023; Xie et al., 2023) have explored volumetric NeRF compression techniques. Re:NeRF (Deng & Tartaglione, 2023) and HollowNeRF (Xie et al., 2023) focuses on voxel pruning, while VQRF (Li et al., 2023) introduces a pruning strategy and compresses voxel features using vector quantization. Rho et al. (Rho et al., 2023) employ wavelet transform to learn a more compact representation. These algorithms utilize techniques in both model compression (*i.e.*, pruning and weight quantization) and data compression (*i.e.*, vector quantization and wavelet transform),

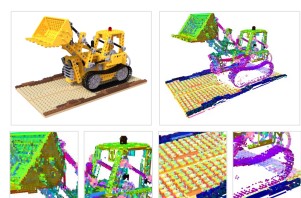

Figure 1: Latent features of explicit neural 3D representation.

and demonstrate promising compression performance. However, these algorithms primarily focus on the voxel representation and overlook the redundancy inherent in the 3D geometry structure and textured latent features. As depicted in Fig. 1, it is clear that latent features follow a certain distribution closely associated with the original color, revealing spatial redundancy between features. This observation suggests the potential for explicit NeRF model compression. Moreover, existing

methods have primarily emphasized compression performance and have not adequately considered the compression time budget, which is crucial for real-world applications.

In this paper, we first conduct a comprehensive analysis of the task of explicit NeRF compression and reformulate it as a 3D data compression problem. Next, we propose a novel NeRF compression framework, named ACRF, which utilizes 3D data compression techniques. The proposed framework focuses on compressing the explicit neural representation, by first pruning the neural 3D structure to points with features, followed by importance-guided feature encoding of latent features. Finally, we apply entropy coding to further compress the encoded results with an importance-prioritized entropy model. To balance both compression performance and encoding time budget, we present two pipelines, ACRF and ACRF-F. By simply combining the rendering and bitrate loss, our compression pipeline, ACRF, can achieve a superior compression performance. By directly distribution fitting, our ACRF-F can achieve fast NeRF compression. Our main contributions can be listed as follows:

- Based on an in-depth analysis of explicit NeRF models, we reformulate the task of NeRF compression, shifting the focus of previous works from model compression to 3D data compression. In light of this reformulation, we present an innovative NeRF compression framework that incorporates advanced 3D data compression techniques.
- We introduce a time-efficient pruning strategy with only view-independent information, and propose two importance-based components: importance-guided feature encoding and importance-prioritized entropy modeling leveraging importance as auxiliary information.
- Our framework is compatible with typical explicit representations (voxel grid and point cloud), and strikes a well balance between rate-distortion and encoding time. Experimental results on several datasets show the great potential and promising scalability of our method.

## 2 RELATED WORK

### 2.1 NEURAL RADIANCE FIELDS

Neural radiance field (NeRF) (Mildenhall et al., 2020) has seen significant improvements in the last three years for novel view synthesis. Follow-up algorithms (Liu et al., 2020; Hedman et al., 2021; Yu et al., 2021; 2022; Sun et al., 2022; Chen et al., 2022a) have introduced volumetric representations to improve inference speed and rendering quality. For instance, NSVF (Liu et al., 2020) develops a sparse voxel octree to represent the 3D scene, while DVGO (Sun et al., 2022) separately models two voxel grids for density and color, and TensorRF (Chen et al., 2022a) decomposes its feature voxel into low-rank tensor components. Apart from the aforementioned voxel-based representations, MobileNeRF (Chen et al., 2023) represents the scene as a triangle mesh textured by deep features. Ost et al. (Ost et al., 2022) merge measured LiDAR point clouds directly into their framework, and PointNeRF (Xu et al., 2022) generates neural points using multi-view stereo and their point growing and pruning strategy. Further, SPIDR (Liang et al., 2022) extends points to SDF.

Recently, a handful of works (Deng & Tartaglione, 2023; Li et al., 2023; Rho et al., 2023; Xie et al., 2023; Shin & Park, 2023; Wang et al., 2023) have been proposed for volumetric NeRF compression. For example, Re:NeRF (Deng & Tartaglione, 2023) and HollowNeRF (Xie et al., 2023) presents a voxel pruning algorithm that iteratively removes voxels. VQRF (Li et al., 2023) proposes a volumetric NeRF compression framework that includes voxel pruning, vector quantization, and post-processing. Rho et al. (Rho et al., 2023) propose to learn a compact representation using wavelet coefficients with a voxel mask.

### 2.2 3D DATA COMPRESSION

Existing algorithms (Meagher, 1982; Schwarz et al., 2018) usually adopt octree structures to encode 3D positions of point clouds or voxel occupancy. For instance, G-PCC (the MPEG point cloud compression standard) (Schwarz et al., 2018), employs an octree-based method for position compression. Learning-based methods (Huang et al., 2020; Biswas et al., 2020; Que et al., 2021; Fu et al., 2022; Chen et al., 2022b) have also leveraged the octree representation and introduced octree-based entropy models for entropy coding.

To achieve attribute compression (*i.e.,* compression of point attributes such as color and reflectance), several techniques have been proposed to extend image compression methods (Wallace, 1992;

Goyal, 2001; Skodras et al., 2001; Sullivan et al., 2012; Bellard, 2015) to point clouds. Generally, this process involves three steps: transform coding, quantization, and entropy coding. Transform coding involves carefully designing a transformation of attributes into the frequency domain to reduce signal redundancy. For example, Zhang et al. (Zhang et al., 2014) construct a graph by dividing a point cloud into blocks and applying the graph Fourier transform to point attributes. Queiroz et al. (De Queiroz & Chou, 2016) propose a region adaptive hierarchical transform (RAHT) by introducing wavelet transforms to point clouds. Quantization is used to quantize coefficients from transform coding into transmitted symbols. Entropy coding aims to encode these symbols into a bitstream through an entropy coder, such as Huffman (Huffman, 1952), arithmetic (Witten et al., 1987), or Golomb-Rice coders (Weinberger et al., 2000; Richardson, 2004). To achieve attribute compression, 3DAC (Fang et al., 2022) introduces an attribute-oriented deep entropy model based on the RAHT tree structure. Rho et al. (Rho et al., 2023) and ReRF (Wang et al., 2023) also use run-length coding and Huffman coding in their NeRF compression pipeline.

## 3 REVISIT NERF COMPRESSION

Neural Radiance Fields (NeRFs) (Mildenhall et al., 2020) leverage implicit neural representation to model a 3D scene through volume rendering. Specifically, it first employs multi-layer perceptrons (MLPs) to map a spatial position $\boldsymbol{x}$ and a view direction $\boldsymbol{d}$ to the point density $\sigma$ and color $\boldsymbol{c}$:

$$(\boldsymbol{c}, \sigma) = \text{MLPs}(\boldsymbol{x}, \boldsymbol{d}). \tag{1}$$

Then, volume rendering (Kajiya & Von Herzen, 1984) is further adopted to accumulate the point densities $\sigma$ and colors $\boldsymbol{c}$ into an expected color $C(\mathbf{r})$ of ray $\mathbf{r}$ following Max (Max, 1995):

$$\widehat{C}(\mathbf{r}) = \sum_{i=1}^{N} T_i \cdot \alpha_i \cdot \boldsymbol{c}_i, \quad \text{where } T_i = \exp\left(-\sum_{j=1}^{i-1} \sigma_j \delta_j\right), \quad \alpha_i = 1 - \exp(-\sigma_i \delta_i). \tag{2}$$

$N$ is the number of sampled points and $\delta_i$ is the distance between two nearby points. Finally, the ray colors are merged into an image with a rendering loss as the optimization objective.

Explicit 3D representations, such as voxel grids (Sun et al., 2022), point clouds (Xu et al., 2022), and meshes (Chen et al., 2023), can be incorporated into the NeRF pipeline to improve rendering quality and training/inference speed (*e.g.*, use spatial interpolation to obtain latent features instead of using MLPs in Eq. (1)). However, these 3D representations also result in a significant increase in model size, which poses a challenge for transmitting and storing these NeRF models.

### 3.1 COMPRESS EXPLICIT NERF AS 3D DATA

The task of NeRF compression is still in its early stages, and there is currently no general and universal framework for this task. Several recent works have investigated this problem from different aspects. A handful of recent studies designed to learn a compact neural representation (Rho et al., 2023; Wang et al., 2023; Shin & Park, 2023), while others focus on model or data compression techniques such as pruning and quantization (Deng & Tartaglione, 2023; Li et al., 2023; Xie et al., 2023). In this section, we first analyze the components of explicit NeRF models and then present our formulation of this task.

First, we identify that explicit NeRF algorithm with additional 3D data structures such as voxel grids, neural points, or meshes in addition to MLPs, require more storage than the original MLP-based NeRF model (5 MB). For example, DVGO (Sun et al., 2022) constructs their model using two voxel grids (100 MB) and a shallow MLP (0.1 MB), while PointNeRF (Xu et al., 2022) combines neural points (over 100 MB) and MLPs (1.4 MB). As shown in Fig. 2, explicit 3D representations take up a significant amount of storage compared to other network parameters (*i.e.*, MLPs).

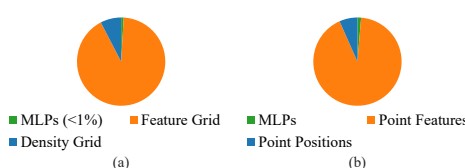

MLPs (<1%)  ■ Feature Grid  ■ MLPs  ■ Point Features
■ Density Grid  ■ Point Positions
(a)                                    (b)

Figure 2: Percentage of disk storage of each part for (a) DVGO (Sun et al., 2022) and (b) Point-NeRF (Xu et al., 2022).

Considering the spatial redundancy illustrated in Fig. 1, we point out that the neural 3D representation constitutes a novel category of 3D data. Thus, we propose to reformulate the problem of explicit

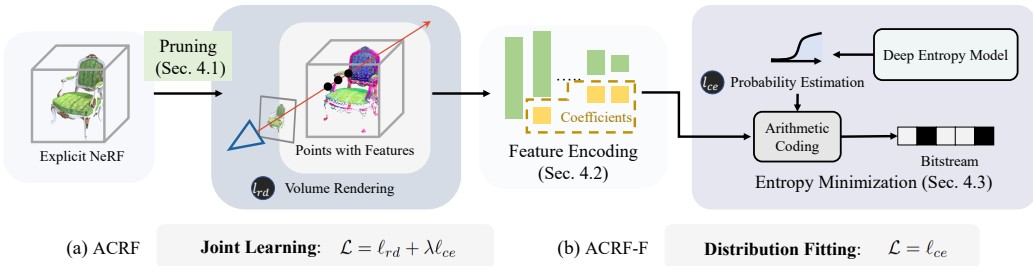

Figure 3: The frameworks of our NeRF compression method, (a) ACRF and (b) ACRF-F.

NeRF compression as the task of **3D data compression**. In other words, we aim to compress the neural 3D representations (including the geometry structure and the textured deep features) to reduce the model size. Given that deep features contribute significantly more information to the NeRF model and consequently require greater storage, we focus on the feature compression stage, also known as attribute compression in the data compression community.

## 4 METHODOLOGY

In this study, we introduce a novel NeRF compression framework, with a particular focus on compressing explicit neural 3D representations. Given a pre-trained NeRF model with an explicit 3D representation, the objective of our framework is to reduce the model size while maintaining high-quality rendering. In particular, we present two compression pipelines, **ACRF** (a full model) and **ACRF-F** (a light model), balancing compression performance and encoding time complexity.

**Compression Pipeline of ACRF.** Figure 3 illustrates the compression framework of ACRF. Given a NeRF model with an explicit neural representation, our framework begins by converting the explicit representation into points with features through view-dependent pruning (VD pruning, Section 4.1). We then perform importance-guided feature encoding for point features (Section 4.2). Furthermore, we minimize the information entropy within the 3D model through jointly training an importance-prioritized entropy model and updating the point features with both cross-entropy (*i.e.,* loss for probability estimation) and rendering loss (Section 4.3). Finally, we convert transform coefficients to a bitstream using arithmetic coding.

**Compression Pipeline of ACRF-F.** The overall pipeline of ACRF-F, as depicted in Fig. 3 (b), is closely similar to the ACRF process. The primary difference lies in the utilization of our proposed view-independent pruning (VI pruning) for the fast removal of less important voxels (Section 4.1). Additionally, we learn the entropy model to fit the coefficient distribution while keeping the point features and other NeRF parameters fixed (Section 4.3).

### 4.1 PRUNING

Voxel pruning (Li et al., 2023; Deng & Tartaglione, 2023; Xie et al., 2023) is a key step in volumetric NeRF compression to remove voxels with low information. Most existing algorithms (Deng & Tartaglione, 2023; Xie et al., 2023) integrate pruning strategies into their training pipeline, which is time-consuming and impractical for compression. In contrast, VQRF (Li et al., 2023) achieves voxel pruning by constructing importance for each voxel and then removing voxels with low cumulative importance.

Here, we refer to the pruning strategy proposed by VQRF as view-dependent pruning as it relies on training camera poses to generate importance (Details can be found in appendix Sec. B.). This approach proves effective because it directly considers blending weights $w_i$ in the rendering function. However, this approach necessitates the availability of trained camera poses for generating the transmittance $T_i$ and weights $w_i$. Additionally, the time required for pruning increases with the number of training views and sampled ray points, making it impractical for several real-world scans (*e.g.*, the Tanks&Temples dataset), where pruning times are even comparable to fine-tuning times

as shown in Table 1. Here, we introduce our **view-independent pruning** strategy leveraging voxel features and sampled points in voxel space.

**Pruning with Voxel Features.** Our approach leverages the insight that the feature tensor $\mathbf{f}_v$ of a voxel grid $\mathbf{v}_l$ encodes essential information for recovering the color and density of the 3D scan. Therefore, statistical measures, such as the maximum absolute value, can be employed to quantify the amount of information contained in a feature tensor. Specifically, we can compute the importance $I_v$ directly from the voxel $v_l$ as the maximum absolute value of the feature $\mathbf{f}_v$:

$$I_v = MAX(|\mathbf{f}_v|). \tag{3}$$

The rationale for utilizing the maximum absolute value of features is grounded in the observation that this value is indicative of the corresponding feature's information content. During initialization, the NeRF model is information-devoid, and most works (*e.g.*, DVGO (Sun et al., 2022) and PointNeRF (Xu et al., 2022)) initialize features with zeros or small random values. During optimization, these features become informational as they encode color and density. As shown in Fig. 1, the feature distribution aligns with the scene over time. More explicitly, the feature values elevate from their origin (typically zeros during initialization) to form a distribution reflecting the scene. Consequently, features with small values approximating zero often contain less information, indicating minimal updates during optimization. In contrast, features with larger values are usually more informative. More statistical analyses about the maximum absolute value are provided in appendix Sec. G.

Our goal for compression is to eliminate features with low informational content and retain those with higher value. This process justifies the use of the distance from the origin as an importance metric for the features. Similar to some model pruning algorithms (Li et al., 2016) which use statistical measures to estimate the importance of convolution kernels, several values, such as the maximum absolute value, the sum of absolute value, and the mean square value, were considered. However, given that features are further fed into NeRF MLPs to generate colors, and networks are usually more sensitive to larger inputs, we opted for the maximum absolute value as the importance indicator. This approach, for instance, would favor preserving a feature of $(16, 0, ..., 0)$ over $(1, 1, ..., 1)$, despite them having the same mean square distance to the origin.

**Pruning with Sampled Points.** In addition to voxel features $\mathbf{f}_v$, the interpolated feature tensor $\mathbf{f}_i$ at the sampled position $\mathbf{x}_i$ and the corresponding probability of termination $\alpha_i$ can also provide important information for neural rendering. Similar to Eq. (3), the sampled point importance $I_x$ can be calculated as $I_x = MAX(|\mathbf{f}_i|) \cdot \alpha_i$. In practice, we adopt a uniform sampling strategy in the 3D space with a sampling step set as half of the grid size. Similar to VQRF (appendix Eq. (1)), this enables us to obtain the importance $I_p$ of the voxel $\mathbf{v}_l$ by evaluating the contributions of sampled points to the voxel's feature tensor. We can finally get our view-independent importance $I_l = I_v \cdot I_p$, then use it to prune voxels through the quantile function in VQRF (appendix Eq. (2)).

## 4.2 FEATURE ENCODING

Following the pruning process, we obtain a concise point representation that includes both point positions (approximately 10%) and point features (about 90%), as shown in Fig. 2 (b). Inspired by previous work (Schnabel & Klein, 2006) which organizes point cloud as octree to skip empty space, we employ octree coding to encode point positions, and then conduct feature encoding.

To reduce point feature redundancy, we employ a point-based wavelet transform, region adaptive hierarchical transform (RAHT) (De Queiroz & Chou, 2016) as the basis for our baseline algorithm. This involves taking quantized points $\mathbf{p}$ from octree coding and converting their features $\mathbf{f}$ into transform coefficients $\mathbf{e}$ as $\mathbf{e} = RAHT(\mathbf{p}, \mathbf{f})$. Notably, RAHT primarily relies on matrix multiplication, allowing us to create a fully differentiable pipeline for feature encoding and subsequent entropy minimization. Details about position encoding and RAHT can be found in appendix Sec. C and D.

**Importance-Guided Feature Encoding.** As illustrated in Sec. 4.1, points with higher importance typically carry more information, and are thus more essential for the accurate recovery of the 3D scan. However, regarding that RAHT is originally designed to process point cloud attributes, primarily focusing on the spatial relationships between point features while neglecting the concept of point importance $I_l$. One straightforward implementation for this limitation is to enhance features based on their importance as $\mathbf{e} = RAHT(\mathbf{p}, \mathbf{f} \cdot I_l)$. More details are provided in appendix Sec. H. Although this approach preserves high-importance features, it introduces additional high-frequency

noise to the original features and necessitates the transmission of importance values for decoding. Consequently, this augmentation significantly increases the final storage requirements.

To deal with these issues, we employ the importance metric $I_l$ as selective masks to split points to several independent point sets, and derive coefficients $\mathbf{e}$ following this formulation:

$$\mathbf{e} = \left\{ \frac{\mathbf{e}_{I_l \in (\theta_0, \theta_1)}}{Q_1}, \frac{\mathbf{e}_{I_l \in (\theta_1, \theta_2)}}{Q_2}, \ldots, \frac{\mathbf{e}_{I_l \in (\theta_{n-1}, \theta_n)}}{Q_n} \right\},$$

$$\mathbf{e}_{I_l \in (\theta_i - 1, \theta_i)} = RAHT(\mathbf{p}_{I_l \in (\theta_i - 1, \theta_i)}, \mathbf{f}_{I_l \in (\theta_i - 1, \theta_i)}). \tag{4}$$

Here, $I_l \in (\theta_i - 1, \theta_i)$ is the selective mask, and $i \in \{0, 1, \ldots, n\}$. We can tune the masks through two sets of hyperparameters, the importance threshold $\theta_i$ and the quantization factor $Q_i$, to encode features with their importance. In practice, $\theta_i$ can be tuned by the quantile function similar to VQRF (appendix Eq. (2)), and $Q_i$ can be set based on the chosen $\theta_i$. Furthermore, the masks can be merged and transmitted using standard zip compression techniques, incurring negligible storage overhead.

### 4.3 ENTROPY MINIMIZATION

According to information theory (Shannon, 1948), the information entropy $H(\mathbf{e})$ of coefficients $\mathbf{e}$ indicates the average level of 'information' as $H(\mathbf{e}) = \mathbb{E}\left[-\log_2 p(\mathbf{e})\right]$, where $p(\mathbf{e})$ is the probability distribution of coefficients $\mathbf{e}$. The objective of this section is to minimize the information entropy $H(\mathbf{e})$ inherent to the NeRF model through imposing constraints on the distribution $p(\mathbf{e})$, thereby reducing the model size of NeRF. Inspired by deep image compression (Ballé et al., 2017), we learn an estimated distribution $q(\mathbf{e})$ to constrain $p(\mathbf{e})$. It is also worth noting that there exists a strong interrelation between the distribution of point features and their corresponding importance $I_l$. Thus, we further introduce $I_l$ as a prior information component.

**Formulation.** Given coefficients $\mathbf{e}$ and importance $I_l$, we aim to minimize the KL Divergence $D$ between the coefficient distribution $p(\mathbf{e})$ and the estimated distribution $q(\mathbf{e})$, incorporating $I_l$ as prior knowledge: $D = \mathbb{E}_{\mathbf{e} \sim p}\left[-\log_2 q\left(\mathbf{e} \mid I_l\right)\right]$. Inspired by deep image compression (Ballé et al., 2017), we learn a deep entropy model to generate the estimated distribution $q(\mathbf{e})$. Moreover, we can further factorize $q(\mathbf{e})$ into a product of conditional probabilities: $q(\mathbf{e}) = \prod_j q\left(e^{(j)} \mid I_l, \mathbf{w}\right)$, where $\mathbf{w}$ is the weights of our entropy model.

**Importance-Prioritized Entropy Modelling.** In the context of our task, the challenge is how to leverage importance $I_l$ as prior information. This is because there is no straightforward one-to-one correspondence between coefficients and importance (The RAHT algorithm has converted the spatial-domain features into the frequency-domain coefficients, and the importance is still in the spatial domain. More details are provided in appendix Sec. D.). One simple yet effective approach is to model multiple independent distributions based on importance $I_l$. Specifically, we reuse the definitions of selective masks and coefficients in Eq. (4): $\mathbf{e} = \{\mathbf{e}_1, \ldots, \mathbf{e}_n\}$, where $\mathbf{e}_i = \frac{\mathbf{e}_{I_l \in (\theta_i - 1, \theta_i)}}{Q_i}$. For each set of coefficients $\mathbf{e}_i$, we introduce independent learnable parameters $\mathbf{w}_i$ to generate the corresponding distribution $q(\mathbf{e}_i)$. Consequently, we can further factorize $q(\mathbf{e})$ as $q(\mathbf{e}) = \prod_i q\left(\mathbf{e}_i \mid \mathbf{w}_i\right) = \prod_{i,j} q\left(e_i^{(j)} \mid \mathbf{w}_i\right)$ for each coefficient $e_i^{(j)}$, and optimize all parameters with a cross-entropy loss $\ell_{ce} = -\sum_{i,j} \log q\left(e_i^{(j)} \mid \mathbf{w}_i\right)$. This approach allows us to implicitly incorporate importance information and thus benefiting entropy modelling.

**Joint Learning.** In our full model ACRF, we simultaneously update the weights of the entropy model and fine-tune the NeRF model. As a result, the cross-entropy loss $\ell_{ce}$ can further reduce the spatial redundancy by updating the transform coefficients. The final optimization loss $\mathcal{L}$ is a combination of the rendering loss $\ell_{rd}$ and the cross-entropy loss $\ell_{ce}$: $\mathcal{L} = \ell_{rd} + \lambda \ell_{ce}$, where $\lambda$ can be tuned to control the model size. Since the cross-entropy loss updates a large number of features, we add $\ell_{ce}$ every 100 iterations to accelerate learning. For the fast mode ACRF, we simply fix the NeRF model and only enable the cross-entropy loss: $\mathcal{L} = \ell_{ce}$.

**Monte Carlo Sampling for Distribution Fitting.** Directly predicting probabilities for all coefficients can be time-consuming, particularly when the number of coefficients is large. For example, PointNeRF (Xu et al., 2022) generates millions of points for scans in the Tanks&Temples dataset, and processing coefficients from these points directly can be challenging. To tackle this issue, we perform random sampling to obtain $n\%$ of coefficients, and estimate the probabilities of only these

coefficients as $\ell_{ce} = -\sum_{i,j \in \mathcal{N}} \log q\left(e_i^{(j)} \mid \mathbf{w}_i\right)$, where $\mathcal{N}$ is the set of sampled indices in each iteration. We set the number of iterations to 100 and the sampling percentage to 1% empirically.

Finally, we have transform coefficients from Sec. 4.2 and predicted distributions of coefficients from our deep entropy model. Thus, the coefficients can be losslessly encoded into a final bitstream through the arithmetic coder. We transmit several components, including a position bitstream, a feature bitstream, the importance masks, the learned entropy model, NeRF MLPs and other metadata. After decoding, we can reconstruct our explicit NeRF model on the decoder side.

## 5 EXPERIMENTS

**(1) Datasets.** We conduct experiments on two datasets:

- **Synthetic-NeRF (Mildenhall et al., 2020)**. This is a view synthesis dataset consisting of 8 synthetic scans, with 100 views used for training and 200 views for testing.
- **Tanks&Temples (Knapitsch et al., 2017)**. This is a real-world dataset with 5 scenes (Barn, Caterpillar, Family, Ignatius, and Truck). We used the default settings of DVGO (Sun et al., 2022) and PointNeRF (Xu et al., 2022) for the voxel and point-based methods, respectively.

**(2) Baselines.** We compare our ACRF and ACRF-F with the following baselines. Additional implementation details and hyperparameter settings are provided in appendix Sec. F.

- **DVGO (Sun et al., 2022)**. This is an explicit NeRF algorithm that uses voxel grids as the neural representation. We train a DVGO model from scratch following the default setting, and use the same model for all voxel-based compression methods to ensure a fair comparison.
- **PointNeRF (Xu et al., 2022)**. This is an explicit NeRF algorithm that uses a point representation. We use the PointNeRF model trained with 200K iterations for compression and adopt the official pre-trained model for all point-based methods.
- **VQRF (Li et al., 2023)**. This is a recent volumetric NeRF compression method. We re-implement the algorithm on the pretrained DVGO model following the official implementation to ensure a fair comparison with our method.
- **VQRF-F**. We directly disable the joint finetune step of VQRF for fast NeRF compression.

### 5.1 EXPERIMENTAL RESULTS

**Quantitative Evaluation.** The quantitative compression results of different methods are presented in Fig. 4. The first row presents a comparison of all voxel-based methods. To obtain the rate-distortion curve for ACRF, we fixed the quantization factors of RAHT and tuned the loss factor $\lambda$, while for ACRF-F, we tuned the quantization factors. All compression algorithms exhibit high compression ratios, highlighting the substantial redundancy in voxel-based representations. Moreover, our proposed methods, ACRF and ACRF-F, outperform VQRF and VQRF-F, respectively. This can

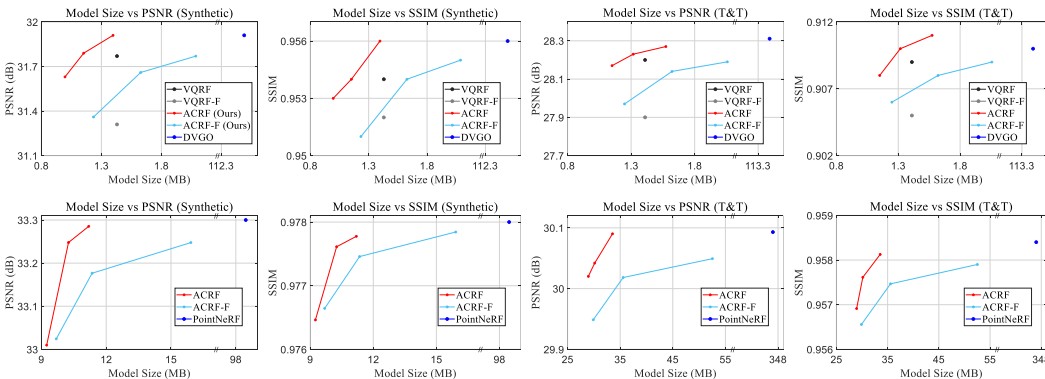

Figure 4: Quantitative results of our ACRF and other NeRF compression approaches on Synthetic-NeRF (Synthetic) and Tanks&Temples (T&T).

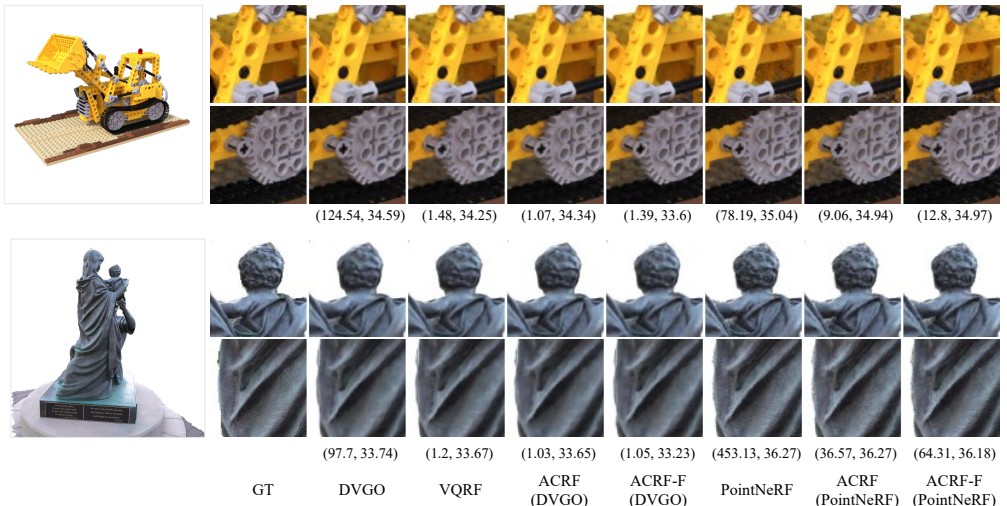

| | | | | | | | |
|---|---|---|---|---|---|---|---|
| | (124.54, 34.59) | (1.48, 34.25) | (1.07, 34.34) | (1.39, 33.6) | (78.19, 35.04) | (9.06, 34.94) | (12.8, 34.97) |
| | (97.7, 33.74) | (1.2, 33.67) | (1.03, 33.65) | (1.05, 33.23) | (453.13, 36.27) | (36.57, 36.27) | (64.31, 36.18) |
| GT | DVGO | VQRF | ACRF (DVGO) | ACRF-F (DVGO) | PointNeRF | ACRF (PointNeRF) | ACRF-F (PointNeRF) |

Figure 5: Qualitative results (*lego* and *Family*) of our ACRF and other baselines. ACRF/ACRF-F (DVGO/PointNeRF) means DVGO/PointNeRF-based ACRF/ACRF-F. Model size in MB and PSNR are displayed below the figure as (model size, PSNR).

| Methods | Synthetic-NeRF | | | | Tanks&Temples | | | |
|---|---|---|---|---|---|---|---|---|
| | VD Pruning | VQ | Finetune | Total | VD Pruning | VQ | Finetune | Total |
| VQRF (Li et al., 2023) | 26s | 40s | 3min 22s | 4min 28s | 4min 31s | 39s | 4min 3s | 9min 14s |
| VQRF-F | 26s | 40s | - | 1min 6s | 4min 31s | 39s | - | 5min 10s |
| | VD Pruning | RAHT | EM | Total | VD Pruning | RAHT | EM | Total |
| ACRF | 26s | 0.5s | 3min 28s | 3min 55s | 4min 31s | 0.4s | 4min 25s | 8min 56s |
| | VI Pruning | RAHT | EM | Total | VI Pruning | RAHT | EM | Total |
| ACRF-F | 2.2s | 0.4s | 1.3s | 3.9s | 2.3s | 0.4s | 1.2s | 3.9s |

Table 1: Encoding time of VQRF and our ACRF. "VQ" and "EM" denotes vector quantization and entropy minimization, respecitvely. "VD" and "VI" denotes view-dependent and view-independent pruning, respectively.

be primarily attributed to the proposed feature and entropy encoding algorithms. The second row presents the results of all point-based methods. For ACRF, our method can achieve a $10 \times$ compression ratio with negligible performance degradation. Compared to the voxel representation, neural points are more compact, making it more challenging to achieve a $100 \times$ compression ratio.

**Qualitative Comparison.** The qualitative results depicted in Fig. 5 are consistent with the quantitative results. Despite a minor reduction in performance, it is difficult to discern any noticeable differences between the rendering outputs of the various algorithms.

**Encoding Time Comparison.** We also compare the encoding time of DVGO-based approaches in Table 1. For consistency and fairness across all experiments, we utilize a workstation equipped with an Intel Xeon Silver 4210 CPU @2.20 GHz and an NVIDIA TITAN RTX GPU. For VQRF and ACRF, we set the same iteration (10K) for finetuning and joint learning (the main part of EM). Our ACRF takes shorter to complete the joint learning phase and achieves a better compression performance than VQRF, because it adopts our feature encoding strategy and learns an auxiliary cross-entropy loss for entropy minimization. Moreover, with our pruning, feature encoding, and entropy coding modules, our ACRF-F can encode a NeRF model within seconds while delivering competitive compression results. During decoding, our methods require additional time for the inverse RAHT transform and entropy decoding, which takes less than 1s.

## 5.2 ABLATION STUDY

We conduct several ablation studies on *chair* of the Synthetic-NeRF dataset.

**View-Independent Pruning.** We investigate the impact of view-independent pruning on our DVGO-based ACRF algorithm. Our experiments, reported in Table 2, show that progressively incorporating the probability of termination (**A**), features (**F**) of sampled points, and features (**F**) of the voxel grid leads to steady improvements in PSNR and reductions in model size. We observe that further incorporating the probability of termination $\alpha$ (**A**) of the voxel grid can result in unacceptable distortion. This is because the density and feature grid of DVGO are separated, and a voxel with low $\alpha$ may also contain valuable features.

| Sampled Points | | Voxel Grid | | PSNR | SIZE |
|---|---|---|---|---|---|
| A | F | A | F | (dB)↑ | (MB)↓ |
| ✓ | | | | 33.97 | 2.46 |
| ✓ | ✓ | | | 33.99 | 2.34 |
| ✓ | ✓ | | ✓ | 33.99 | 2.18 |
| ✓ | ✓ | ✓ | ✓ | 31.08 | 1.16 |

Table 2: Ablation study on the view-independent voxel pruning (Sec. 4.1). "A" and "F" denote the probability of termination $\alpha$ and features, respectively.

**Importance-based Feature Encoding and Entropy Modelling.** We further study the effect of importance-guided feature encoding and importance-prioritized entropy modelling on our ACRF and ACRF-F. As depicted in Fig. 6 (a), we start with two baseline models without our importance-based strategies, denoted as 'baseline A' and 'baseline B' for ACRF-F and ACRF, respectively. Subsequently, we introduce importance-guided feature encoding (Imp FE) to these two baseline models. It is worth noting that directly applying feature encoding with importance masks might initially lead to a performance decrease. This is primarily because transforms in different regions yield coefficients that follow distinct distributions. We further include the importance-prioritized entropy modelling to develop our ACRF and ACRF-F. The results clearly demonstrate that the combination of importance-based feature encoding and entropy modeling significantly improves the compression performance of both ACRF and ACRF-F, which underscores the effectiveness of our importance-based strategies in enhancing the overall performance of our methods.

**Effectiveness of Components.** As shown in Fig. 6 (b), we evaluate the effectiveness of each component of our framework. We start with the original DVGO method and progressively incorporate our view-independent pruning, deep entropy model, RAHT, joint learning, and importance-based modules (importance-guided feature encoding and importance-prioritized entropy modelling). Finally, we can obtain our full model, ACRF. The results show that the performance is consistently improved with the addition of each module, which proves the effectiveness of each component.

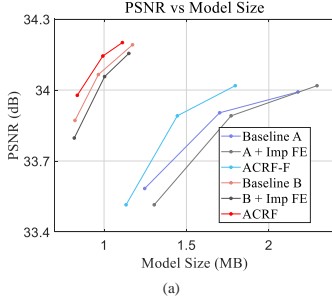 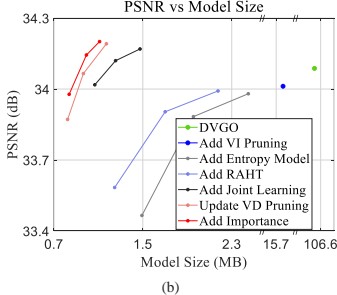

Figure 6: Ablation studies. (a): Ablation study on the importance-based feature and entropy encoding. (b): Ablation study on each component of our framework.

## 6 CONCLUSION

In this paper, we propose an explicit NeRF compression algorithm. Our method includes several key components, including a view-independent pruning strategy, importance-guided feature encoding, and importance-prioritized entropy modeling. Extensive experimental results validate the effectiveness of our 3D-data-oriented NeRF compression methodology, showcasing a remarkable improvement in compression performance while achieving a reduction in encoding time complexity. It is feasible to further extend our algorithm for dynamic and real-time NeRF compression.

**Acknowledgements.** This work was partially supported by the National Natural Science Foundation of China (No. U20A20185).

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
