# APPENDIX – ACRF: COMPRESSING EXPLICIT NEURAL RADIANCE FIELDS VIA ATTRIBUTE COMPRESSION

**Guangchi Fang[1], Qingyong Hu[2], Longguang Wang[3], Yulan Guo[4]**
[1]Sun Yat-sen University, [2]University of Oxford, [3]Aviation University of Air Force,
[4]National University of Defense Technology

## A VISUALIZATION OF LATENT FEATURES

Figure 1 shows more visualization results of latent features. We project features from latent space to the color space through t-SNE, and the local patterns indicate the spatial feature redundancy.

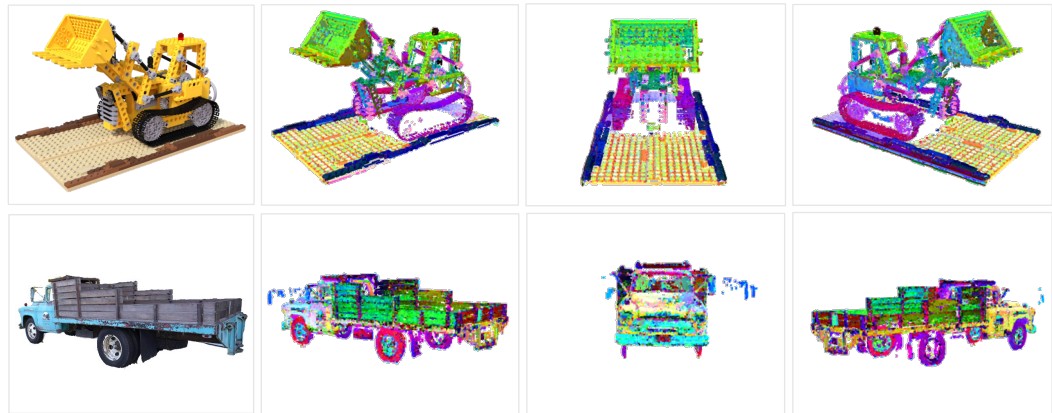

Figure 1: More visualization results of latent features in color space.

## B DETAILS OF VIEW-DEPENDENT PRUNING

We refer to the pruning strategy proposed by VQRF (Li et al., 2023) as view-dependent pruning as it relies on training camera poses to generate voxel importance for pruning.

Specifically, VQRF assigns blending weights $w_i = T_i \cdot \alpha_i$ to sampled ray points $\mathbf{x}_i$ as the importance $I_i$, and obtains the importance $I_l$ of voxel $\mathbf{v}_l$ by aggregating the importance of neighbor points $\mathcal{N}_l$ around $\mathbf{v}_l$, depending on their distance:

$$I_l = \sum_{\mathbf{x}_i \in \mathcal{N}_l} (1 - |\mathbf{v}_l - \mathbf{x}_i|) \cdot I_i, \quad \text{where } |\mathbf{v}_l - \mathbf{x}_i| \leq 1. \tag{1}$$

The algorithm further prunes voxels below the importance threshold $\theta$ as:

$$\theta = F^{-1}(\beta), \quad F(\theta) = \frac{\sum I_l \cdot \mathbf{1}\{I_l < \theta\}}{\sum I_l}, \tag{2}$$

where the function $F$ denotes the cumulative score rate with $\mathbf{1}\cdot$ representing the binary indicator, and $\beta$ is a hyperparameter that indicates the total amount of importance to be pruned.

## C DETAILS OF POSITION ENCODING

Following the pruning process, we obtain a more concise point representation that includes both point positions (approximately 10%) and point features (about 90%). Inspired by previous work

(Schnabel & Klein, 2006) which organizes point cloud as octree to skip empty space, we employ octree coding[1] to encode point positions.

We conduct experiments on the Synthetic-NeRF dataset to evaluate position encoding. For the DVGO-based methods, we losslessly compress the occupancy mask of feature voxels with octree coding. Following the default setting of DVGO, the voxel size is around 200 and the occupancy mask is 61 KB. The file size of the mask is reduced to 21 KB after encoding, and the time for encoding and decoding is 0.7 s and 0.2 s, respectively. For the PointNeRF-based method, we encode the point positions using a 12-depth-level octree (*i.e.*, 4096 resolution). The file size is reduced from 5.8 MB to 0.8 MB, with PSNR being slightly decreased from 33.30 dB to 33.27 dB. The encoding and decoding time is 4.7 s and 2.6 s, respectively. Recent well-optimized implementation of octree coding can be implemented in real time (within 0.1 s) (Huang et al., 2020; Biswas et al., 2020; Que et al., 2021), which further shows the potential of our method.

## D  DETAILS OF REGION ADAPTIVE HIERARCHICAL TRANSFORM

Different from VQRF (Li et al., 2023) that quantizes features in latent space, our feature encoding aims to remove the spatial redundancy between point features. As shown in Fig. 1, we project point features from latent space to the color space through t-SNE. It is clear that the colors (*i.e.*, latent features) exhibit a certain spatial distribution, indicating a high level of spatial redundancy between point features.

**Region Adaptive Hierarchical Transform.** To reduce the spatial redundancy of point features, we employ region adaptive hierarchical transform (RAHT) (De Queiroz & Chou, 2016), a point-based wavelet transform that is commonly used for processing point attributes such as colors. In this work, we leverage RAHT to encode point features. Specifically, we use the quantized points with features and convert the features into frequency-domain coefficients. As shown in Fig. 2, the subscripts of $l_{d,x,y}$ denote depth, $x$, and $y$, re-

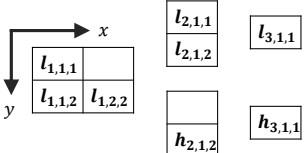

Figure 2:  A 2D example of Region Adaptive Hierarchical Transform.

spectively. First, feature tensors $l_{1,1,1}$, $l_{1,1,2}$, and $l_{1,2,2}$ are transformed along the $x$ axis. $l_{1,1,1}$ is directly transformed to $l_{2,1,1}$ at the next level as $l_{d+1,x,y} = l_{d,x,y}$, and two neighbors, $l_{1,1,2}$ and $l_{1,2,2}$, are converted to low- and high-frequency coefficients, $l_{2,1,2}$ and $h_{2,1,2}$, as:

$$\begin{bmatrix} l_{d+1,x,y} \\ h_{d+1,x,y} \end{bmatrix} = \mathbf{T}_{w_1,w_2} \begin{bmatrix} l_{d,x,y} \\ l_{d,x+1,y} \end{bmatrix}, \quad \mathbf{T}_{w_1,w_2} = \frac{1}{\sqrt{w_1+w_2}} \begin{bmatrix} \sqrt{w_1} & \sqrt{w_2} \\ -\sqrt{w_2} & \sqrt{w_1} \end{bmatrix}, \quad (3)$$

where $\mathbf{T}_{w_1,w_2}$ is the transform matrix, and $w_1$ and $w_2$ are the weights (*i.e.*, number of points at the corresponding space) of $l_{d,x,y}$ and $l_{d,x+1,y}$, respectively. The final DC coefficient $l_{3,1,1}$ and all high-frequency coefficients $h$ should be compressed and transmitted to the decoder side, and the inverse RAHT transform can reconstruct all feature tensors. RAHT converts the spatial-domain features into the frequency-domain coefficients to reduce low-frequency information. However, the importance obtained by both view-dependent and independent pruning is still in the spatial domain. Thus, there is no straightforward one-to-one correspondence between coefficients and importance.

By applying uniform quantization to the transform coefficients (Ballé et al., 2017), feature encoding can be performed in a differentiable manner as the primary operation of RAHT is matrix multiplication (Eq. (3)). Therefore, it is feasible to impose a bitrate constraint on these coefficients to regulate the storage size of the original feature tensors through optimization.

## E  DISCUSSION OF POINT-BASED ALGORITHM

As shown in Fig. 3, we evaluate point pruning and importance-based modules on our PointNeRF-based method (ACRF-F). Considering that PointNeRF (Xu et al., 2022) assigns each point with a confidence score for pruning, we directly adopt the learned confidence as the importance measure and prune around 10% points in this experiment (VQRF (Li et al., 2023) prunes around 90% voxels).

---

[1]We use octree coding with entropy coding in the MPEG reference implementation: https://github.com/MPEGGroup/mpeg-pcc-tmc13.

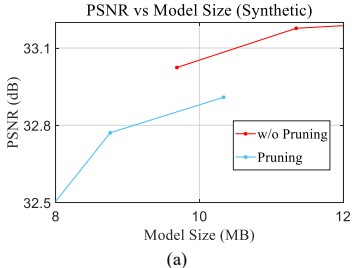 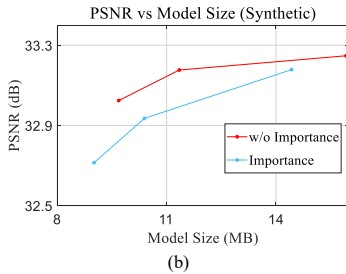

(a)                                           (b)

Figure 3: Ablation study of point pruning and importance-based modules.

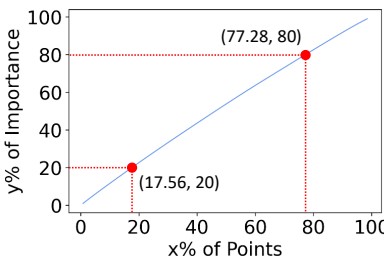

Figure 4: We draw the quantile-quantile curve following (Li et al., 2023), which means x% of least important points contributes to y% percent of total importance. In contrast to the voxel-based method (Li et al., 2023), the point-based method contains points with similar importance.

In Fig. 3 (a), we tune the uniform quantization step of feature coding to obtain two rate-distortion curves, and the only difference between them is enabling pruning or not. The results indicate that the further pruning might lead to a performance drop on both Synthetic-NeRF. In Fig. 3 (b), we include our importance-guided feature encoding and importance-prioritized entropy model. The results also show a performance drop, especially at a low bitrate.

To further investigate this problem, we analyze the importance of each point in Fig. 4. We provide the quantile-quantile curve following (Li et al., 2023), which means x% of least important points contributes to y% percent of total importance. We note that, in contrast to feature voxels, the neural points are more compact and share similar importance scores. Consequently, we do not merge point pruning and importance-based modules for our point-based NeRF framework and leave it as a future problem.

## F  IMPLEMENTATION DETAILS

We conduct experiments for both voxel- and point-based NeRF models (DVGO and PointNeRF).

For ACRF-F (DVGO), we set pruning quantile $\beta_p = 0.9999$ for voxel pruning (prune less important voxels contributing $0.01\%$ importance). We set the importance threshold $\theta_i$, according to importance quantile $\beta_{imp;i}$, where $\beta_{imp;0} = 0$, $\beta_{imp;1} = 0.99$ and $\beta_{imp;2} = \beta_p = 0.9999$. The corresponding quantization factors are set as $Q_1 = 1 \cdot \lambda_Q$ and $Q_2 = 10 \cdot \lambda_Q$. Thus, we can simply tune the factor $\lambda_Q$ to obtain the rate-distortion curve of ACRF-F.

For ACRF (DVGO), we set $\beta_p = 0.999$, $\beta_{imp;0} = 0$, $\beta_{imp;1} = 0.99$ and $\beta_{imp;2} = \beta_p = 0.999$. We also set $Q_1 = 1 \cdot \lambda_Q$ and $Q_2 = 10 \cdot \lambda_Q$ but fix $\lambda_Q$ as 0.5. The loss factor $\lambda$ is tuned from 0.002 to 0.01 to obtain the rate-distortion curve of ACRF.

For ACRF-F (PointNeRF) and ACRF (PointNeRF), we simply disable the pruning and importance-based modules, and employ the basic framework for our experiments.

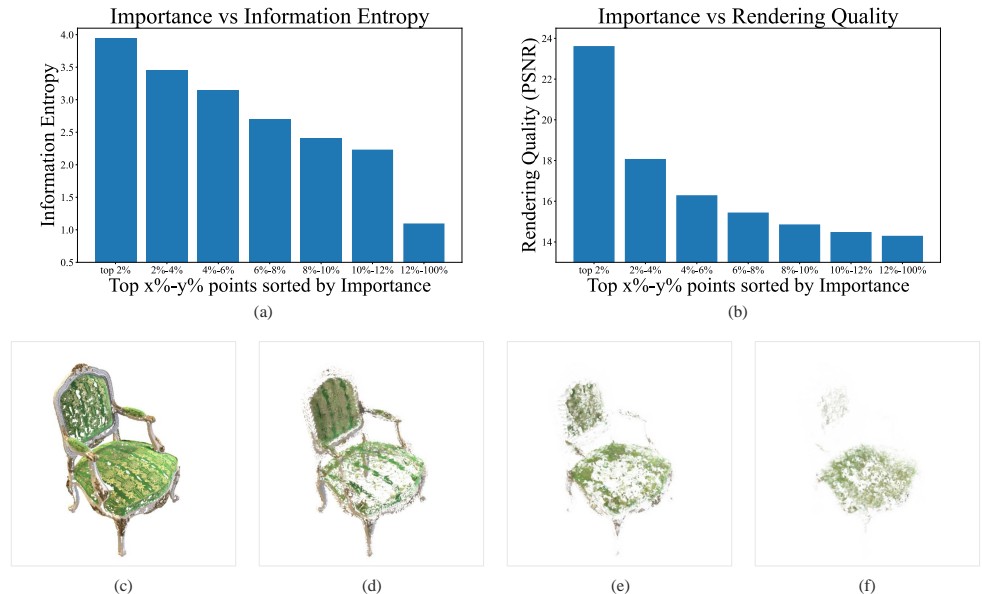

Figure 5: Statistical analysis of view-independent importance. (a): The relationship between importance and information entropy. (b): The relationship between importance and rendering quality. (c) to (f): Rendering results with top 2%, top 2%-4%, top 4%-6% and top 6%-8%, respectively.

## G    STATISTICAL EVIDENCE FOR MAXIMUM ABSOLUTE VALUE

We provide more statistical analyses of our proposed view-independent importance in Fig. 5. Notably, the importance is calculated based on the maximum absolute value of the feature, following Sec. 4.1. Our objective is to quantify the amount of information through (a) information entropy, indicating the average level of "information", and (b) rendering quality, which directly reflects the performance of a NeRF model. Subsequently, we conduct statistical analyses on these metrics with our importance measure, aiming to unveil the relationship between our importance (the maximum absolute value) and the amount of information.

In Fig. 5 (a), we illustrate the relationship between our importance $I_l$ and the information entropy $H(\mathbf{f}_v)$ of gird feature $\mathbf{f}_v$. We sort all grid points based on their importance and select top $x\%$ to $y\%$ points with features. To further obtain their information entropy $H(\mathbf{f}_v)$, we discretize the continuous variable $\mathbf{f}_v$ into a discrete variable $\hat{\mathbf{f}}_v$ and calculate information entropy as:

$$H(\mathbf{f}_v) \approx H(\hat{\mathbf{f}}_v) = - \sum_{\hat{f}_v \in \hat{\mathbf{f}}_v} p(\hat{f}_v) \log p(\hat{f}_v), \quad \text{where } \hat{\mathbf{f}}_v = round(\mathbf{f}_v). \tag{4}$$

The results, as shown in Fig. 5 (a), demonstrate that point features with higher importance contain larger information entropy. This observation indicates that the maximum absolute value can effectively serve as a measure to quantify the amount of information.

In Fig. 5 (b), we depict the relationship between our importance and rendering quality. Following the sorting of all grid points based on their importance, we retain the top $x\%$ to $y\%$ points with features. The results highlight that points with higher importance play a more crucial role in maintaining the performance of a NeRF model. Additionally, qualitative rendering results of the top 2% to top 6%-8% points are provided in Fig. 5 (c) to (f). These qualitative results align with the quantitative findings in Fig. 5 (b), reinforcing that the amount of information can be effectively indicated by our importance measure based on the maximum absolute value.

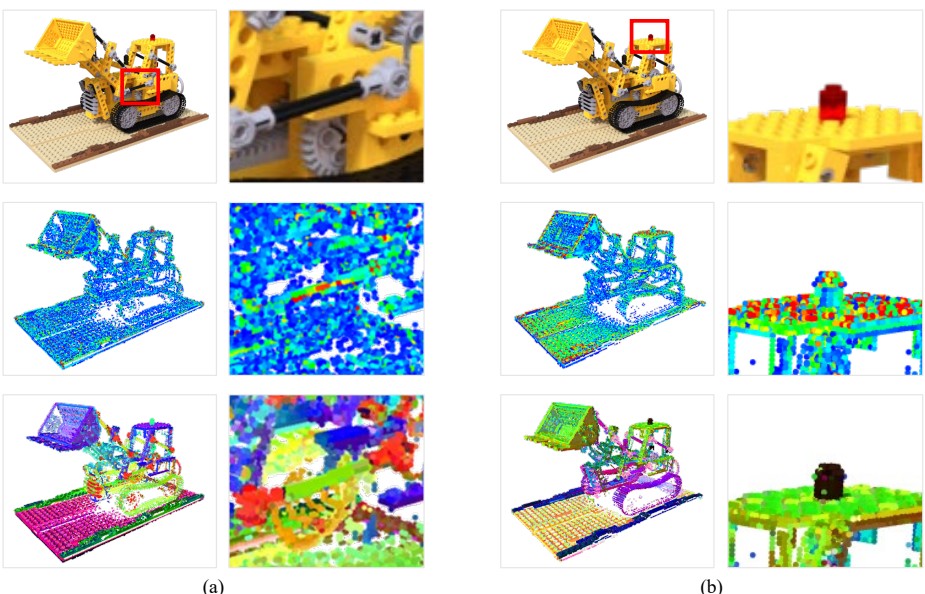

(a)                                                                            (b)

Figure 6: (a): Visualization of the GT image (first row), view-independent importance (second row) and the top 1% point features sorted by view-independent importance (last row). (b): Visualization of the GT image, view-dependent importance and the corresponding top 1% point features.

# H STRAIGHTFORWARD IMPLEMENTATION OF IMPORTANCE-GUIDED FEATURE ENCODING

Given point positions $\mathbf{p}$ and features $\mathbf{f}$, we transform features into coefficients through RAHT transform as $\mathbf{e} = RAHT(\mathbf{p}, \mathbf{f})$. Details about RAHT are provided in appendix Sec. D.

Incorporating importance $I_l$ as prior information, our objective is to leverage $I_l$ to effectively preserve point features with heightened significance. In the initial implementation described in Sec. 4.2, we obtain coefficients as:

$$\mathbf{e} = RAHT(\mathbf{p}, \mathbf{f}_{I_l}), \quad \mathbf{f}_{I_l} = \mathbf{f} \cdot I_l. \tag{5}$$

Throughout the decoding process, the reconstructed features $\hat{\mathbf{f}}$ can be recovered from the inverse RAHT transform while considering importance:

$$\hat{\mathbf{f}} = \frac{\hat{\mathbf{f}}_{I_l}}{I_l}, \quad \hat{\mathbf{f}}_{I_l} = iRAHT(\mathbf{p}, \mathbf{e}). \tag{6}$$

Note that, for the recovery of point features, the transmission of importance values is requisite for decoding.

We further provide an examination of the high-frequency noise issue. As shown in Fig. 6 (a) and (b), we visualize the GT image, view-independent and view-independent importance, and the top 1% point features sorted by their corresponding importance. Importance is visualized as a heatmap, where red signifies higher values and blue lower ones, and features are projected into color space through t-SNE. In Fig. 6 (a), the point feature representation of the black stick of the lego in the GT image appears as green in the projected feature image. However, in the importance figure, it is composed of red, yellow, and blue elements. A similar observation is also shown in Fig. 6 (b). This illustrates that, unlike point features, point importance lacks a robust correlation with the scene color. Consequently, the direct combination of features with importance may introduce additional noise to the original features.

## I ABLATION STUDIES ON SYNTHETIC-NERF

Ablation studies on the whole Synthetic-NeRF dataset are provided in Fig. 7, and these findings are consistent with the results obtained for the chair.

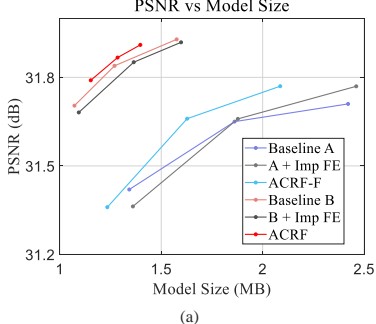
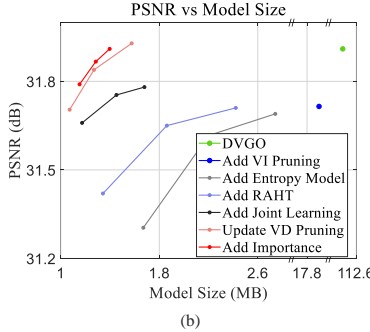

Figure 7: Ablation studies on Synthetic-NeRF. (a): Ablation study on the importance-based feature and entropy encoding. (b): Ablation study on each component of our framework.

## J COMPARISON WITH OTHER METHODS

Additional quantitative compression results for different methods are presented in Fig. 8. We compare our ACRF and ACRF-F with VQRF (Li et al., 2023), Re:NeRF (Deng & Tartaglione, 2023) and Rho et al. (Rho et al., 2023). Notably, ACRF, ACRF-F, VQRF and Re:NeRF are based on DVGO, while Rho et al. is built upon TensoRF.

Figure 8 (a) shows comparisons on the Synthetic-NeRF dataset. It is clear that our ACRF outperforms other DVGO-based algorithms. Note that Rho et al. attains the highest rendering result due to its original model, TensoRF-VM-384, achieving a PSNR of 33.21 dB on Synthetic-NeRF, while DVGO only gets 31.91 dB. Despite a relatively large performance drop in PSNR, Rho et al. still gets the best quantitative results.

Figure 8 (b) shows comparisons on the Tanks&Temples dataset. In this more complex real-world dataset, our ACRF outperforms all other algorithms. Although TensoRF still achieves a better rendering quality than DVGO (28.56 dB versus 28.31 dB in PSNR), the relatively large distortion introduced by Rho et al. places this TensoRF-based algorithm behind both ACRF and VQRF. Notably, our lightweight model, ACRF-F, achieves a similar compression result to Rho et al, while our ACRF-F requires only a few seconds for encoding, while Rho et al. demands 24 mins for training (see Table 1 in appendix of Rho et al.).

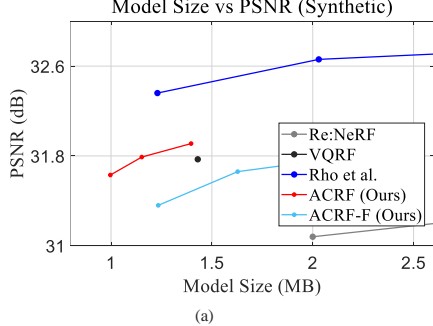
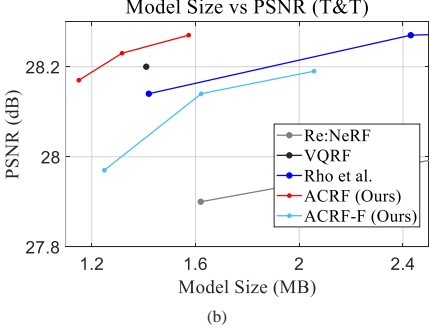

Figure 8: Quantitative results of our ACRF and other NeRF compression approaches on Synthetic-NeRF (Synthetic) and Tanks&Temples (T&T).

## K  ADDITIONAL EXPERIMENTAL RESULTS

We provide additional quantitative (Tables 1, 3, 2 and 4) and qualitative (Figs. 9 and 10) results to show our compression performance. DVGO-Q and PointNeRF-Q are obtained by direct weight quantization of features from *float64* to *int8*. The results on these datasets are consistent with those provided in our paper, which shows the effectiveness of our proposed method.

| | Method | Chair | Drums | Ficus | Hotdog | Lego | Materials | Mic | Ship | Avg. |
|---|---|---|---|---|---|---|---|---|---|---|
| | | | | | **Synthetic-NeRF** | | | | | |
| Size(MB) (↓) | DVGO | 106.54 | 95.81 | 109.86 | 133.80 | 124.54 | 176.07 | 49.87 | 103.42 | 112.49 |
| | DVGO-Q | 16.60 | 14.43 | 17.67 | 21.53 | 20.37 | 26.32 | 6.81 | 17.71 | 17.68 |
| | VQRF (DVGO) | 0.99 | 0.90 | 1.06 | 1.46 | 1.48 | 3.10 | 0.41 | 2.06 | 1.43 |
| | ACRF (DVGO) | 0.84 | 0.98 | 0.98 | 0.85 | 1.07 | 2.60 | 0.37 | 1.54 | 1.15 |
| | VQRF-F (DVGO) | 0.99 | 0.90 | 1.06 | 1.46 | 1.48 | 3.10 | 0.41 | 2.06 | 1.43 |
| | ACRF-F (DVGO) | 1.13 | 0.84 | 0.89 | 1.21 | 1.39 | 2.41 | 0.39 | 1.61 | 1.23 |
| PSNR(dB) (↑) | DVGO | 34.09 | 25.47 | 32.68 | 36.66 | 34.59 | 29.51 | 33.15 | 29.12 | 31.91 |
| | DVGO-Q | 33.68 | 25.39 | 32.64 | 36.60 | 34.53 | 29.51 | 31.89 | 28.94 | 31.65 |
| | VQRF (DVGO) | 33.78 | 25.37 | 32.69 | 36.51 | 34.25 | 29.28 | 33.15 | 29.16 | 31.77 |
| | ACRF (DVGO) | 33.98 | 25.46 | 32.69 | 36.12 | 34.34 | 29.41 | 33.12 | 29.20 | 31.79 |
| | VQRF-F (DVGO) | 32.94 | 25.11 | 32.33 | 36.03 | 33.41 | 29.09 | 32.63 | 28.92 | 31.31 |
| | ACRF-F (DVGO) | 33.51 | 25.28 | 32.37 | 35.72 | 33.60 | 29.21 | 32.56 | 28.65 | 31.36 |
| SSIM(↑) | DVGO | 0.976 | 0.930 | 0.978 | 0.980 | 0.976 | 0.950 | 0.983 | 0.878 | 0.956 |
| | DVGO-Q | 0.974 | 0.929 | 0.978 | 0.980 | 0.976 | 0.950 | 0.977 | 0.877 | 0.955 |
| | VQRF (DVGO) | 0.974 | 0.927 | 0.977 | 0.978 | 0.973 | 0.945 | 0.982 | 0.877 | 0.954 |
| | ACRF (DVGO) | 0.975 | 0.930 | 0.977 | 0.975 | 0.973 | 0.947 | 0.982 | 0.877 | 0.954 |
| | VQRF-F (DVGO) | 0.969 | 0.924 | 0.976 | 0.976 | 0.970 | 0.944 | 0.980 | 0.874 | 0.952 |
| | ACRF-F (DVGO) | 0.972 | 0.925 | 0.976 | 0.971 | 0.970 | 0.942 | 0.978 | 0.871 | 0.951 |
| $LPIPS_{ALEX}$(↓) | DVGO | 0.027 | 0.078 | 0.025 | 0.034 | 0.027 | 0.059 | 0.018 | 0.160 | 0.054 |
| | DVGO-Q | 0.029 | 0.080 | 0.025 | 0.034 | 0.027 | 0.060 | 0.024 | 0.161 | 0.055 |
| | VQRF (DVGO) | 0.032 | 0.083 | 0.028 | 0.039 | 0.030 | 0.066 | 0.020 | 0.160 | 0.057 |
| | ACRF (DVGO) | 0.030 | 0.080 | 0.027 | 0.053 | 0.033 | 0.065 | 0.021 | 0.165 | 0.059 |
| | VQRF-F (DVGO) | 0.037 | 0.088 | 0.029 | 0.043 | 0.035 | 0.068 | 0.023 | 0.165 | 0.061 |
| | ACRF-F (DVGO) | 0.037 | 0.088 | 0.029 | 0.063 | 0.037 | 0.073 | 0.029 | 0.172 | 0.066 |
| $LPIPS_{VGG}$(↓) | DVGO | 0.017 | 0.060 | 0.015 | 0.018 | 0.013 | 0.027 | 0.014 | 0.116 | 0.035 |
| | DVGO-Q | 0.019 | 0.062 | 0.015 | 0.018 | 0.014 | 0.028 | 0.023 | 0.118 | 0.037 |
| | VQRF (DVGO) | 0.018 | 0.061 | 0.017 | 0.019 | 0.013 | 0.033 | 0.014 | 0.113 | 0.036 |
| | ACRF (DVGO) | 0.017 | 0.059 | 0.017 | 0.026 | 0.015 | 0.031 | 0.014 | 0.119 | 0.037 |
| | VQRF-F (DVGO) | 0.023 | 0.066 | 0.017 | 0.021 | 0.017 | 0.034 | 0.017 | 0.121 | 0.039 |
| | ACRF-F (DVGO) | 0.021 | 0.066 | 0.017 | 0.031 | 0.017 | 0.039 | 0.020 | 0.127 | 0.042 |

Table 1: Per-scene results on Synthetic-NeRF of the DVGO-based methods.

| | Method | Chair | Drums | Ficus | Hotdog | Lego | Materials | Mic | Ship | Avg. |
|---|---|---|---|---|---|---|---|---|---|---|
| | | | | | **Synthetic-NeRF** | | | | | |
| Size(MB) (↓) | PointNeRF | 111.79 | 87.21 | 64.69 | 65.39 | 78.19 | 191.42 | 82.10 | 106.92 | 98.46 |
| | PointNeRF-Q | 27.23 | 21.81 | 16.58 | 16.48 | 19.66 | 46.46 | 20.59 | 26.23 | 24.38 |
| | ACRF | 11.82 | 10.53 | 10.54 | 6.69 | 9.06 | 19.47 | 9.39 | 12.06 | 11.19 |
| | ACRF-F | 17.54 | 14.79 | 12.95 | 10.01 | 12.80 | 28.73 | 13.31 | 17.24 | 15.92 |
| PSNR(dB) (↑) | PointNeRF | 35.40 | 26.08 | 36.13 | 37.25 | 35.04 | 29.61 | 35.95 | 30.97 | 33.30 |
| | PointNeRF-Q | 35.39 | 26.07 | 36.12 | 37.25 | 35.04 | 29.60 | 35.94 | 30.97 | 33.30 |
| | ACRF | 35.49 | 26.05 | 36.34 | 37.20 | 34.94 | 29.57 | 35.78 | 30.92 | 33.29 |
| | ACRF-F | 35.33 | 26.05 | 36.10 | 37.19 | 34.97 | 29.58 | 35.86 | 30.90 | 33.25 |
| SSIM(↑) | PointNeRF | 0.991 | 0.954 | 0.993 | 0.991 | 0.988 | 0.971 | 0.994 | 0.942 | 0.978 |
| | PointNeRF-Q | 0.991 | 0.954 | 0.993 | 0.991 | 0.988 | 0.971 | 0.994 | 0.942 | 0.978 |
| | ACRF | 0.991 | 0.954 | 0.993 | 0.990 | 0.988 | 0.970 | 0.993 | 0.941 | 0.978 |
| | ACRF-F | 0.991 | 0.954 | 0.993 | 0.991 | 0.988 | 0.971 | 0.994 | 0.941 | 0.978 |
| $LPIPS_{ALEX}$(↓) | PointNeRF | 0.010 | 0.054 | 0.009 | 0.016 | 0.011 | 0.041 | 0.007 | 0.070 | 0.027 |
| | PointNeRF-Q | 0.010 | 0.054 | 0.009 | 0.016 | 0.011 | 0.041 | 0.007 | 0.070 | 0.027 |
| | ACRF | 0.010 | 0.054 | 0.009 | 0.018 | 0.011 | 0.044 | 0.008 | 0.072 | 0.028 |
| | ACRF-F | 0.010 | 0.054 | 0.009 | 0.017 | 0.011 | 0.042 | 0.008 | 0.071 | 0.028 |
| $LPIPS_{VGG}$(↓) | PointNeRF | 0.023 | 0.076 | 0.022 | 0.037 | 0.024 | 0.072 | 0.014 | 0.124 | 0.049 |
| | PointNeRF-Q | 0.023 | 0.076 | 0.022 | 0.038 | 0.024 | 0.072 | 0.014 | 0.124 | 0.049 |
| | ACRF | 0.024 | 0.077 | 0.021 | 0.042 | 0.025 | 0.074 | 0.015 | 0.127 | 0.051 |
| | ACRF-F | 0.024 | 0.077 | 0.022 | 0.039 | 0.024 | 0.073 | 0.015 | 0.125 | 0.050 |

Table 2: Per-scene results on Synthetic-NeRF of the PointNeRF-based methods.

| | **Tanks&Temples** | | | | | |
| --- | --- | --- | --- | --- | --- | --- |
| | Method | Barn | Caterpillar | Family | Ignatius | Truck | **Avg.** |
| Size(MB) (↓) | DVGO | 137.69 | 116.76 | 97.70 | 102.08 | 112.73 | 113.39 |
| | DVGO-Q | 21.01 | 17.60 | 15.28 | 14.85 | 17.61 | 17.27 |
| | VQRF (DVGO) | 1.81 | 1.46 | 1.20 | 1.12 | 1.43 | 1.41 |
| | ACRF (DVGO) | 1.64 | 1.50 | 1.03 | 0.98 | 1.44 | 1.32 |
| | VQRF-F (DVGO) | 1.81 | 1.46 | 1.20 | 1.12 | 1.43 | 1.41 |
| | ACRF-F (DVGO) | 1.63 | 1.30 | 1.05 | 0.98 | 1.29 | 1.25 |
| PSNR(dB) (↑) | DVGO | 26.82 | 25.71 | 33.74 | 28.20 | 27.09 | 28.31 |
| | DVGO-Q | 26.51 | 25.46 | 33.47 | 28.06 | 26.96 | 28.09 |
| | VQRF (DVGO) | 26.72 | 25.55 | 33.67 | 28.07 | 26.97 | 28.20 |
| | ACRF (DVGO) | 26.82 | 25.55 | 33.65 | 28.05 | 27.08 | 28.23 |
| | VQRF-F (DVGO) | 26.35 | 25.24 | 33.24 | 28.05 | 26.61 | 27.90 |
| | ACRF-F (DVGO) | 26.44 | 25.35 | 33.23 | 28.09 | 26.75 | 27.97 |
| SSIM(↑) | DVGO | 0.838 | 0.904 | 0.962 | 0.943 | 0.905 | 0.910 |
| | DVGO-Q | 0.834 | 0.902 | 0.960 | 0.942 | 0.904 | 0.908 |
| | VQRF (DVGO) | 0.838 | 0.901 | 0.961 | 0.941 | 0.903 | 0.909 |
| | ACRF (DVGO) | 0.840 | 0.903 | 0.960 | 0.941 | 0.905 | 0.910 |
| | VQRF-F (DVGO) | 0.830 | 0.898 | 0.958 | 0.941 | 0.898 | 0.905 |
| | ACRF-F (DVGO) | 0.832 | 0.898 | 0.958 | 0.941 | 0.899 | 0.906 |
| $\text{LPIPS}_{ALEX}$(↓) | DVGO | 0.294 | 0.169 | 0.070 | 0.087 | 0.161 | 0.156 |
| | DVGO-Q | 0.299 | 0.173 | 0.071 | 0.090 | 0.162 | 0.159 |
| | VQRF (DVGO) | 0.296 | 0.177 | 0.071 | 0.089 | 0.164 | 0.159 |
| | ACRF (DVGO) | 0.295 | 0.173 | 0.073 | 0.089 | 0.164 | 0.159 |
| | VQRF-F (DVGO) | 0.310 | 0.180 | 0.076 | 0.091 | 0.171 | 0.166 |
| | ACRF-F (DVGO) | 0.307 | 0.181 | 0.076 | 0.092 | 0.170 | 0.165 |
| $\text{LPIPS}_{VGG}$(↓) | DVGO | 0.289 | 0.151 | 0.063 | 0.092 | 0.146 | 0.148 |
| | DVGO-Q | 0.296 | 0.156 | 0.064 | 0.096 | 0.149 | 0.152 |
| | VQRF (DVGO) | 0.285 | 0.154 | 0.060 | 0.091 | 0.144 | 0.147 |
| | ACRF (DVGO) | 0.282 | 0.152 | 0.063 | 0.092 | 0.143 | 0.146 |
| | VQRF-F (DVGO) | 0.310 | 0.164 | 0.066 | 0.096 | 0.156 | 0.158 |
| | ACRF-F (DVGO) | 0.306 | 0.163 | 0.067 | 0.097 | 0.155 | 0.157 |

Table 3: Per-scene results on Tanks&Temples of the DVGO-based methods.

| | **Tanks&Temples** | | | | | |
| --- | --- | --- | --- | --- | --- | --- |
| | Method | Barn | Caterpillar | Family | Ignatius | Truck | **Avg.** |
| Size(MB) (↓) | PointNeRF | 212.68 | 322.19 | 453.13 | 350.36 | 396.74 | 347.02 |
| | PointNeRF-Q | 51.59 | 77.70 | 106.66 | 84.44 | 94.03 | 82.89 |
| | ACRF | 22.56 | 35.12 | 36.57 | 37.75 | 35.72 | 33.54 |
| | ACRF-F | 32.53 | 50.29 | 64.31 | 54.89 | 60.60 | 52.53 |
| PSNR(dB) (↑) | PointNeRF | 29.41 | 27.14 | 36.27 | 29.20 | 28.45 | 30.09 |
| | PointNeRF-Q | 29.41 | 27.14 | 36.27 | 29.20 | 28.44 | 30.09 |
| | ACRF | 29.37 | 27.09 | 36.27 | 29.16 | 28.56 | 30.09 |
| | ACRF-F | 29.35 | 27.11 | 36.18 | 29.19 | 28.42 | 30.05 |
| SSIM(↑) | PointNeRF | 0.940 | 0.941 | 0.989 | 0.967 | 0.955 | 0.958 |
| | PointNeRF-Q | 0.940 | 0.940 | 0.989 | 0.967 | 0.955 | 0.958 |
| | ACRF | 0.939 | 0.940 | 0.989 | 0.967 | 0.955 | 0.958 |
| | ACRF-F | 0.939 | 0.940 | 0.989 | 0.967 | 0.954 | 0.958 |
| $\text{LPIPS}_{ALEX}$(↓) | PointNeRF | 0.125 | 0.100 | 0.016 | 0.060 | 0.070 | 0.074 |
| | PointNeRF-Q | 0.125 | 0.100 | 0.016 | 0.060 | 0.070 | 0.074 |
| | ACRF | 0.126 | 0.102 | 0.017 | 0.060 | 0.070 | 0.075 |
| | ACRF-F | 0.127 | 0.101 | 0.017 | 0.060 | 0.071 | 0.075 |
| $\text{LPIPS}_{VGG}$(↓) | PointNeRF | 0.181 | 0.145 | 0.034 | 0.072 | 0.106 | 0.107 |
| | PointNeRF-Q | 0.181 | 0.145 | 0.034 | 0.072 | 0.106 | 0.107 |
| | ACRF | 0.182 | 0.146 | 0.035 | 0.072 | 0.107 | 0.108 |
| | ACRF-F | 0.183 | 0.146 | 0.035 | 0.072 | 0.107 | 0.109 |

Table 4: Per-scene results on Tanks&Temples of the PointNeRF-based methods.

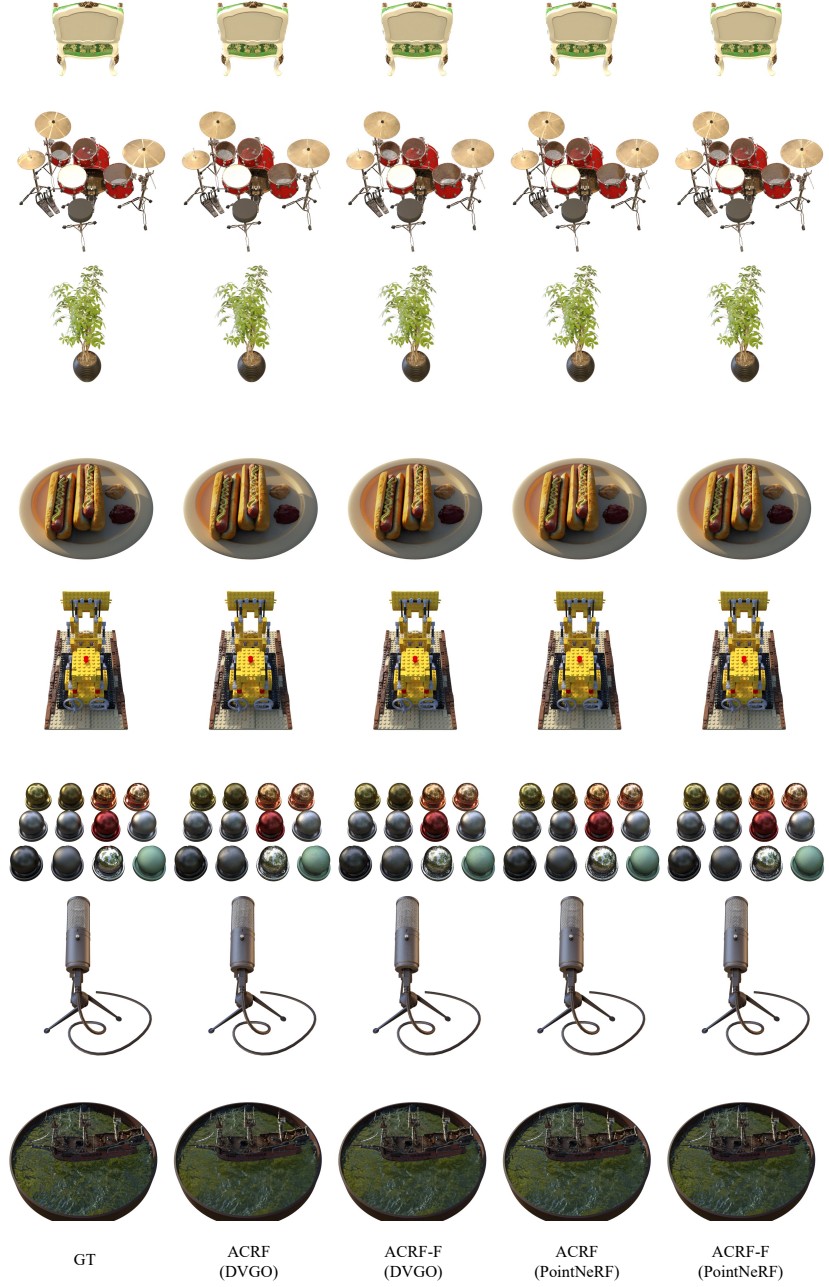

GT     ACRF (DVGO)     ACRF-F (DVGO)     ACRF (PointNeRF)     ACRF-F (PointNeRF)

Figure 9: Qualitative results on Synthetic-NeRF.

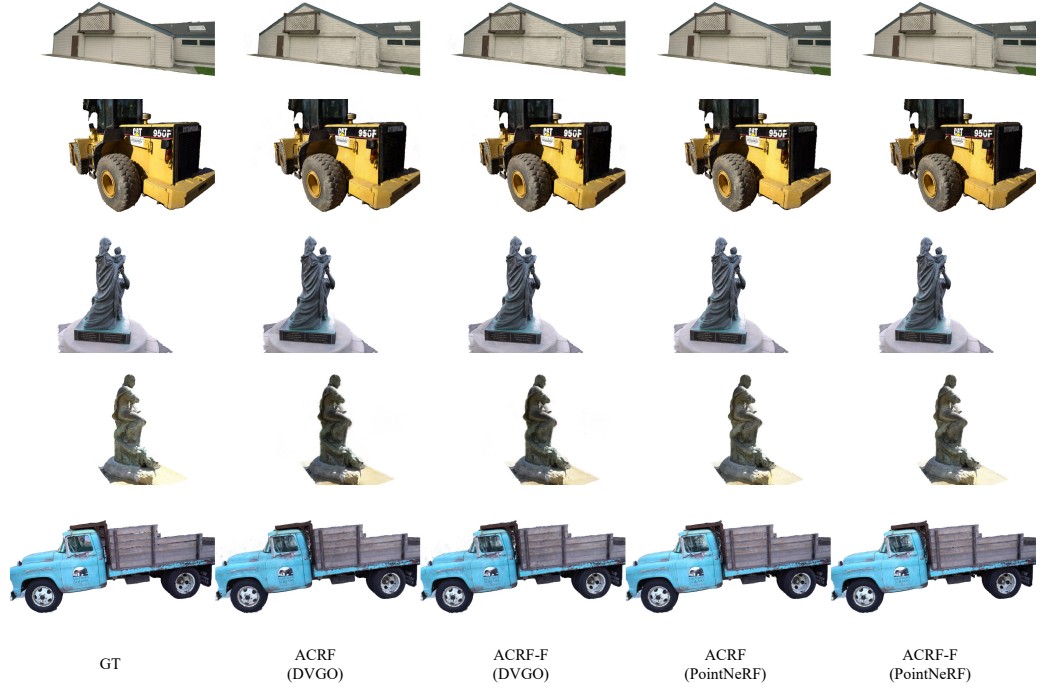

GT ACRF (DVGO) ACRF-F (DVGO) ACRF (PointNeRF) ACRF-F (PointNeRF)

Figure 10: Qualitative results on Tanks&Temples.