# OpenReview forum: "ACRF: Compressing Explicit Neural Radiance Fields via Attribute Compression"
_ICLR.cc/2024/Conference — ICLR 2024 poster_

### Official Review · Reviewer_sVJV · 2023-10-29

**Soundness:** 3 good
**Presentation:** 2 fair
**Contribution:** 3 good
**Rating:** 6
**Confidence:** 4

**Summary:**

Given a pre-trained NeRF model with an explicit 3D representation, this paper aims to reduce the model size while maintaining good performance. It proposes reformulating the task as 3D data compression with three steps.
Firstly, it proposes view-dependent pruning, which eliminates features with low absolute values.
Then, it performs a point-based wavelet transform with octree coding to convert features with importance to coefficients.
Lastly, it minimizes the information entropy for the coefficients.
Depending on whether there is a rendering loss, two versions are introduced.
Experiments are conducted on the Synthetic-NeRF and Tank&Temples datasets to show their significant compression rates.

**Strengths:**

[Results] The quantitive results in Fig. 4 and encoding time comparison Tab. 1 show the proposed method achieves a modestly larger compression ratio in comparison with voxel-based methods and a remarkably higher compression ratio in comparison with point-based methods.

[Novelty] The proposed three-step method is reasonable, achieves remarkable improvements in compression performance, and simultaneously reduces the encoding time.

**Weaknesses:**

[Clarity]

- In Sec.4.1, the paper uses a half page and two formulations to introduce view-dependent pruning. I think this part seems redundant. Because the following view-independent pruning is not based on the view-dependent pruning.

- In Sec.4.2, the paper introduces $e = RAHT(p,f \cdot I_l)$ as a straightforward implementation. I am not sure how to introduce the importance and then recover $f$. Actually, I think $f \cdot I_l $ destroy $f$ and it is hard to recover $f$. It is not a noise issue. Instead, I think the proposed mask strategy is more intuitive. Also, if $e = RAHT(p,f \cdot I_l)$ is important, it would be better to use it as a baseline. I think this part introduces more confusion.

- I could not follow the first formula in Section 4.3, the independent variables are distributions p and q, but in the right of the equation, there is only q. I am not sure how to eliminate the distribution of p. Are any assumptions or references missing?

- Fig.1 is too small, and it does not provide enough explanation. I cannot follow the meaning of the color or the connection between local patterns and latent features.

- Fig.5 is hard to distinguish. All the results of different methods seem the same. It would be better to highlight their difference.

[Fig.4] In Fig.4, the compressed model achieves a 100 $\times$ compression ratio while obtaining a higher SSIM (top right). Would you please provide more discussion on it?

[Speed & Practicality] In Sec.5.1 Encoding Time Comparison part, the paper claims that the proposed method only requires additional 1s for decoding. This part is vague. I am not sure whether this time has a huge impact or is insignificant during testing. The overall compression seems complex. I suggest the decoding time and the overall inference time should be discussed in detail. More specific settings, including batch size, image size, and overall inference time, are needed.

[$|f_v|$] Eq.5 is not so convicting. For me, I think it is easy to understand that the minimum absolute value means low information. But I am not sure the maximum absolute value can be employed to quantify the amount of information. The paper claims this is based on''statistical measures''. It would be better to provide statistical evidence or some references to support this claim.

[Task specific] The feature encoding and the entropy minimization seem like standard strategies for data compression and are not specifically designed for Nerf. It would be better to highlight the main contribution of those two parts regarding Nerf.


[Experiments]

- This paper claims that integrating pruning strategies into training pipeline is time-consuming and impractical for compression(Section 4.1). But it is still important to compare those existing methods(Deng & Tartaglione, 2023; Xie et al., 2023), even the proposed method is a post-optimization method.

- The comparison experiments are insufficient. I suggest to compare previous works like (Li et al.,2023; Rho et al., 2023;)

[Typo]
- Section 1, propsoe -> propose
- Appendix Section C, subscript error
-The description of baselines should be more specific (Fig. 5), like the definitions of "ACRF(DVGO)" and "ACRF-F(DVGO)".

**Questions:**

See weakness

---

> ### Author Response · Authors · 2023-11-22
> **Response to Reviewer sVJV**
>
> Thank you for your time and helpful feedback. We respond below to your questions and concerns. Please note that additional figures are included in the updated version of the appendix (Supplementary Material).
>
> **Q1. Introduction of view-dependent pruning.**
>
> We shrink the introduction of view-dependent pruning in the main paper and move detailed descriptions to the appendix.
>
>
> **Q2. Straightforward implementation of importance-guided feature encoding.**
>
>
> We provide more elaborations about straightforward implementation of importance-guided feature encoding in appendix Sec. H as follows.
>
> Given point positions $\mathbf{p}$ and features $\mathbf{f}$, we transform features into coefficients through RAHT transform as $\mathbf{e}=\text{RAHT}(\mathbf{p}, \mathbf{f})$. Details about RAHT are provided in appendix Sec. D.
>
> Incorporating importance $I_l$ as prior information, our objective is to leverage $I_l$ to effectively preserve point features with heightened significance. In the initial implementation described in Sec. 4.2, we obtain coefficients as:
> $$\mathbf{e}=\text{RAHT}(\mathbf{p}, \mathbf{f}_I), \quad \mathbf{f}_I=\mathbf{f} \cdot I_l.$$Throughout the decoding process, the reconstructed features $\hat{\mathbf{f}}$ can be recovered from the inverse RAHT transform while considering importance:
> $$\hat{\mathbf{f}}=\frac{\hat{\mathbf{f}}_I}{I_l}, \quad \hat{\mathbf{f}}_I=\text{iRAHT}(\mathbf{p}, \mathbf{e}).$$Note that, for the recovery of point features, the transmission of importance values is requisite for decoding.
>
> We further provide an examination of the high-frequency noise issue. As shown in appendix Fig. 6 (a) and (b), we visualize the GT image, view-independent and view-independent importance, and the top 1\% point features sorted by their corresponding importance. Importance is visualized as a heatmap, where red signifies higher values and blue lower ones, and features are projected into color space through t-SNE.
>
>
> In appendix Fig. 6 (a), the point feature representation of the black stick of the lego in the GT image appears as green in the projected feature image. However, in the importance figure, it is composed of red, yellow, and blue elements. A similar observation is also shown in appendix Fig. 6 (b). This illustrates that, unlike point features, point importance lacks a robust correlation with the scene color. Consequently, the direct combination of features with importance may introduce additional noise to the original features.
>
> We agree that the proposed mask strategy is also an intuitive approach. Given that our strategy is directly built upon the original RAHT, we use RAHT as the baseline algorithm.
>
>
> **Q3. The first formula in Section 4.3.**
>
> The first formula in Section 4.3 indicates the Shannon cross entropy. We agree that updating our formula following previous works [1] would be more appropriate, as $D = \mathbb{E}_{\mathbf{e} \sim p}\left[-\log _2 q\left(\mathbf{e} \mid I_l\right)\right]$. This formula is widely employed in image compression, for instance, in equation (7) of [2] and equation (1) of [1].
>
> [1] Variational image compression with a scale hyperprior.
>
> [2] End-to-end optimized image compression.
>
>
>
> **Q4. Explanation of Fig.1.**
>
> To visualize latent features, we employ t-SNE to project high-dimensional features from feature space to a 3-dimensional color space, as depicted in Fig. 1 in the main paper and Sec. A in the appendix. It is clear that latent features follow a certain distribution closely associated with the original color. Moreover, the latent features exhibit clustering, resembling visualization results often observed in scene decomposition tasks. In order words, the local patterns (or patches) of latent features reveal significant spatial redundancy among nearby features.
>
>
> **Q5. Fig.5 is hard to distinguish.**
>
> We highlight the specific model size and quality info in Fig.5 to indicate their difference.
>
>
> **Q6. Discussion on SSIM in Fig.4.**
>
> The main reason of the higher SSIM lies in the joint learning stage of ACRF, which further optimizes the DVGO model. To ensure a fair comparison with VQRF, the optimization iteration is set at 10K, consistent with the finetune steps of VQRF.

---

> > ### Author Response · Authors · 2023-11-22
> > **Response to Reviewer sVJV (2)**
> >
> > **Q7. Speed and Practicality.**
> >
> > During encoding, starting with the feature voxel grid of a pretrained DVGO model, we first conduct pruning to obtain point features. Subsequently, feature encoding is applied to transform these features into coefficients. The following step involves minimizing the entropy of the model and entropy encoding.
> >
> > The decoding process serves as the inverse of encoding and primarily comprises two steps: entropy decoding (approximately 0.5s) and feature decoding (around 0.4s). We initiate entropy decoding to recover coefficients from the bitstream and subsequently reconstruct features from these coefficients through feature decoding. The reconstructed model also includes others components such as voxel positions, MLPs and metadata, which can be directly transmitted or compressed using conventional techniques like zip and octree coding. The process of this part is not the focal point of 3D compression and is designed to be similar to previous works. Upon completion of the decoding stage, the NeRF model is reconstructed. Notably, the only difference between the reconstructed model and the original lies in their voxel feature values. Consequently, both models share the same inference stage, with unchanged inference time.
> >
> > DVGO takes 4mins 31s to render the whole testset (200 images) of 'chair' in the Synthetic-NeRF dataset following the default setting (800*800 resolution) on a workstation equipped with an Intel Xeon Silver 4210 CPU @2.20 GHz and an NVIDIA TITAN RTX GPU. Consequently, the overall inference time amounts 4mins 32s, with the decoding time being nearly negligible.
> >
> >
> > **Q8. Statistical evidence for maximum absolute value.**
> >
> > We provide more analyses about our view-independent importance (maximum absolute value) in appendix Sec. G as follows.
> >
> > We provide more statistical analyses of our proposed view-independent importance in appendix Fig. 5. Notably, the importance is calculated based on the maximum absolute value of the feature, following Sec. 4.1.  Our objective is to quantify the amount of information through (a) information entropy, indicating the average level of "information", and (b) rendering quality, which directly reflects the performance of a NeRF model. Subsequently, we conduct statistical analyses on these metrics with our importance measure, aiming to unveil the relationship between our importance (the maximum absolute value) and the amount of information.
> >
> >
> > In appendix Fig. 5 (a), we illustrate the relationship between our importance $I_l$ and the information entropy $H(\mathbf{f}_v)$ of gird feature $\mathbf{f}_v$. We sort all grid points based on their importance and select top x% to y% points with features. To further obtain their information entropy $H(\mathbf{f}_v)$, we discretize the continuous variable $\mathbf{f}_v$ into a discrete variable $\hat{\mathbf{f}}_v$ and calculate information entropy as:
> > $$H(\mathbf{f}_v) \approx H(\hat{\mathbf{f}}_v) =  -\sum p(\hat{f}_v) \log p(\hat{f}_v), \quad \text{where } \hat{f}_v \in \hat{\mathbf{f}}_v \text{ and } \hat{\mathbf{f}}_v = \text{round}(\mathbf{f}_v).$$The results, as shown in appendix Fig. 5 (a), demonstrate that point features with higher importance contain larger information entropy. This observation indicates that the maximum absolute value can effectively serve as a measure to quantify the amount of information.
> >
> >
> > In appendix Fig. 5 (b), we depict the relationship between our importance and rendering quality. Following the sorting of all grid points based on their importance, we retain the top x% to y% points with features. The results highlight that points with higher importance play a more crucial role in maintaining the performance of a NeRF model. Additionally, qualitative rendering results of the top 2% to top 6%-8% points are provided in appendix Fig. 5 (c) to (f). These qualitative results align with the quantitative findings in appendix Fig. 5 (b), reinforcing that the amount of information can be effectively indicated by our importance measure based on the maximum absolute value.
> >
> >
> >
> > **Q9. Task specific.**
> >
> > We agree that utilizing feature encoding and the entropy minimization for NeRF constitutes one of the main contributions in our work. To the best of our knowledge, our ACRF is the first NeRF compression algorithm focusing on 3D data compression techniques, including feature encoding and entropy encoding.

---

> > > ### Author Response · Authors · 2023-11-22
> > > **Response to Reviewer sVJV (3)**
> > >
> > > **Q10. Comparison with other pruning algorithms.**
> > >
> > > We agree the potential benefits of incorporating a pruning strategy into optimization for NeRF compression. However, in our current task setting, our objective is to propose a universal NeRF compression framework designed for compressing a pre-trained explicit NeRF model. Additionally, as of now, there are no official implementations and reported training times for the works mentioned (Deng \& Tartaglione, 2023; Xie et al., 2023). We leave the exploration of the online pruning strategy as a future study and will update the comparison results in our subsequent work.
> > >
> > >
> > > **Q11. Comparison with other compression methods.**
> > >
> > > We provide more comparison with other compression methods in appendix Sec. J as follows.
> > >
> > > Additional quantitative compression results for different methods are presented in appendix Fig. 8. We compare our ACRF and ACRF-F with VQRF (Li et al.,2023), Re:NeRF (Deng \& Tartaglione, 2023) and Rho et al. (Rho et al., 2023). Notably, ACRF, ACRF-F, VQRF and Re:NeRF are based on DVGO, while Rho et al. is built upon TensoRF.
> > >
> > >
> > > Appendix Fig. 8 (a) shows comparisons on the Synthetic-NeRF dataset. It is clear that our ACRF outperforms other DVGO-based algorithms. Note that Rho et al. attains the highest rendering result due to its original model, TensoRF-VM-384, achieving a PSNR of 33.21 dB on Synthetic-NeRF, while DVGO only gets 31.91 dB. Despite a relatively large performance drop in PSNR, Rho et al. still gets the best quantitative results.
> > >
> > >
> > > Appendix Fig. 8 (b) shows comparisons on the Tanks\&Temples dataset. In this more complex real-world dataset, our ACRF outperforms all other algorithms. Although TensoRF still achieves a better rendering quality than DVGO (28.56 dB versus 28.31 dB in PSNR), the relatively large distortion introduced by Rho et al. places this TensoRF-based algorithm behind both ACRF and VQRF. Notably, our lightweight model, ACRF-F, achieves a similar compression result to Rho et al, while our ACRF-F requires only a few seconds for encoding, while Rho et al. demands 24 mins for training (see Table 1 in appendix of Rho et al.).
> > >
> > >
> > > **Q12. Typo.**
> > >
> > > We thank reviewer for the detailed comment. We have fixed the typo in our revised manuscript.

---

### Official Review · Reviewer_GWvi · 2023-10-31

**Soundness:** 3 good
**Presentation:** 3 good
**Contribution:** 3 good
**Rating:** 8
**Confidence:** 4

**Summary:**

This paper proposes a novel approach for explicit NeRF compression, focusing on compressing the latent features of the NeRF model. More specifically, the authors first conduct a comprehensive analysis of NeRF compression and reformulate the task as 3D data compression. Then, they propose their ACRF framework, comprising pruning, feature encoding, and entropy minimization. Experiments were conducted to demonstrate the effectiveness of the proposed method, both for the compression performance and coding speed.

**Strengths:**

1. Originality: The analysis and reformulation of NeRF compression seems reasonable, and the utilization of attribute compression is practical.
2. Quality: The experimental results validate the effectiveness of the proposed algorithm, which integrates the conventional image compression pipeline with several NeRF-oriented modules.
3. Clarity: The paper is well-written and easy to read.
4. Significance: Given the rising popularity of NeRF models, the investigation of NeRF compression is of great significance.

**Weaknesses:**

1. From the perspective of conventional image compression, additional computation and time budget is required for encoding and decoding, which might be a problem for real NeRF applications.
2. In the related works (Sec. 2.2), it would be better to illustrate the main difference between the proposed method and prior research, such as the mentioned 3DAC, Rho et al. and ReRF.
3. Experiments: In Table 2, it seems that there is a performance drop with additional information (Voxel Grid, A). Please check the result and give some explanations. Similarly, in Figure 6, Baseline A outperforms A+Imp FE. Please justify.
4. Experiments: In Figure 5, the qualitative results of two relatively simple scans are provided. It would be beneficial to extend the analysis to include more complex scans (e.g., object with high reflectance).
5. In the appendix (Sec. D), it seems that the proposed modules do not perform well on PointNeRF. It would be beneficial to extend the analysis in the appendix, and add some discussions in the main paper.

**Questions:**

1. Include discussion and future direction for recent NeRF algorithms. The gaussian-splatting algorithm represents the radiance field in a point-cloud-like structure, showcasing notable achievements in training and rendering performance. One limitation of this approach is the significant storage budget. For example, the output model takes over 1GB for unbounded scenes, like bicycle in mipnerf360. It will be interesting to integrate these techniques with the proposed algorithm.
2. See weakness 3, 4, 5 and 6.

---

> ### Author Response · Authors · 2023-11-22
> **Response to Reviewer GWvi**
>
> Thank you for your time and helpful feedback. We respond below to your questions and concerns. Please note that additional figures are included in the updated version of the appendix (Supplementary Material).
>
>
> **Q1. Additional computation and time budget.**
>
> We agree that the necessity of additional encoding and decoding steps in the current compression pipeline. Similar to image and video compression, NeRF-oriented compression algorithms become necessary for the transmission and storage of NeRF models, especially given the rising popularity of explicit NeRF. Consequently, the additional computation and time budget are inevitable. In this regard, the exploration of real-time NeRF compression stands as a practical research direction, and we leave it as a future study.
>
>
>
> **Q2. Difference between ACRF and prior works.**
>
> 3DAC constructs a RAHT-based entropy model for point cloud attribute compression, Rho et al. introduce a 2D wavelet transform-based algorithm for plane-based NeRF models (TensoRF) and ReRF presents a residual field for dynamic NeRF models.
>
>
> **Q3. Performance drop in Table 2 and Figure 6.**
>
> 1. The observed performance drop with additional information (Voxel Grid, A) can be attributed to the use of two separate grid representations by DVGO. Specifically, DVGO models density and color features with two distinct grids, allowing voxels with low density to still contribute valuable features to sampled ray points through trilinear interpolation.
>
> 2. The observed results in Figure 6 can be attributed to the mixed coefficients distributions generated by transforms in regions with varying importance. A naive entropy model, commonly used in deep image compression, fails to adequately fit this composite distribution during optimization. Consequently, we introduce importance as side information for entropy modelling to develop our full model.
>
>
> **Q4. More qualitative results.**
>
> Additional qualitative results are available in appendix Sec. K, and these results are consistent with those provided in the main paper. Notably, for more complex scans such as object with high reflectance, our algorithm's rendering results resemble the uncompressed model, exhibiting similar artifacts. Addressing this issue may involve enhancing the original NeRF model through the incorporation of advanced reconstruction algorithms, such as BRDF estimation.
>
>
> **Q5. Limited improvement on PointNeRF.**
>
> The main reason is that PointNeRF adopts a more compact point representation. As shown in appendix Figure 4, neural points in PointNeRF exhibit closely similar importance values, which leads to the ineffectiveness of the importance-based modules. Nonetheless, with our basic framework, we achieve a 10:1 compression ratio with negligible performance degradation on PointNeRF.
>
> **Q6. Discussion about gaussian-splatting and future direction.**
>
> Gaussian-splatting has garnered considerable attention recently due to its remarkable rendering performance and fast optimization and inference speed. The memory and storage consumption of this algorithm primarily consist of millions of 3D Gaussians, each containing a 3D position, scales, rotation, opacity, and spherical harmonics. We agree that exploring storage and memory compression for Gaussian-splatting is an interesting research avenue. Addressing this challenge could involve employing model-specific pruning and feature encoding algorithms, and we leave it for future study.

---

### Official Review · Reviewer_HSPa · 2023-11-01

**Soundness:** 4 excellent
**Presentation:** 4 excellent
**Contribution:** 3 good
**Rating:** 8
**Confidence:** 4

**Summary:**

This manuscript proposed a novel framework of Radiance Field attribute compression, which treated the compression task of explicit neural 3D representation as 3D data compression. Specifically, the neural 3D structure is pruned and converted to points with features, which are further encoded using importance-guided feature encoding. An importance-prioritized entropy model is proposed to estimate the probability distribution of transform coefficients, which are then entropy coded with an arithmetic coder using the predicted distribution. Experimental results demonstrate that the proposed method achieves superior performance on both synthetic and real-world datasets such as Synthetic-NeRF and Tanks&Temples.

**Strengths:**

The whole manuscript is well structured and the technique details are easy to follow. Experimental results demonstrate that the proposed method achieves superior performance on both synthetic and real-world datasets, in terms of RD performance and encoding/decoding time,.

The method proposed in this manuscript follows the standard point cloud attribute compression process, and has been optimized based on the characteristics of the explicit neural 3D representation in multiple stages such as data pruning, feature encoding and entropy minimization. The idea is reasonable and interesting.

**Weaknesses:**

1）Some technique details are not clear enough. For example, only encoding time data are provided in Table 1, it is suggested to provide decoding time data to highlight the practicality of the proposed algorithm. The model size and quality info are missing in Fig. 5.

2)  Other type of compression methods such as Rho et al. (Rho et al., 2023) are not evaluated in this manuscript.

3) Typos. ``As depicted in 6,``=>``As depicted in Fig. 6,``

**Questions:**

Please refer to the Weaknesses section.

---

> ### Author Response · Authors · 2023-11-22
> **Response to Reviewer HSPa**
>
> Thank you for your time and helpful feedback. We respond below to your questions and concerns. Please note that additional figures are included in the updated version of the appendix (Supplementary Material).
>
>
> **Q1. Decoding time in Table 1 and model size with quality info in Fig. 5.**
>
>
> 1. The decoding stage comprises two parts, first entropy decoding (around 0.5s) and then transform decoding (around 0.4s). Consequently, the overall decoding time takes under 1s. It is worth noting that these components are currently implemented in Python. However, in practical applications, re-implementation in C++ is recommended [1, 2].
>
>
> 2. Detailed quantitative results are available in appendix (Table 1 to 4). We have incorporated the corresponding results to Fig. 5 in our revised manuscript.
>
> [1] Compression of 3d point clouds using a region-adaptive hierarchical transform
>
> [2] https://github.com/digitalivp/RAHT
>
>
> **Q2. Comparison with other compression methods.**
>
> We provide more comparison with other compression methods in appendix Sec. J as follows.
>
> Additional quantitative compression results for different methods are presented in appendix Fig. 8. We compare our ACRF and ACRF-F with VQRF (Li et al.,2023), Re:NeRF (Deng \& Tartaglione, 2023) and Rho et al. (Rho et al., 2023). Notably, ACRF, ACRF-F, VQRF and Re:NeRF are based on DVGO, while Rho et al. is built upon TensoRF.
>
>
> Appendix Fig. 8 (a) shows comparisons on the Synthetic-NeRF dataset. It is clear that our ACRF outperforms other DVGO-based algorithms. Note that Rho et al. attains the highest rendering result due to its original model, TensoRF-VM-384, achieving a PSNR of 33.21 dB on Synthetic-NeRF, while DVGO only gets 31.91 dB. Despite a relatively large performance drop in PSNR, Rho et al. still gets the best quantitative results.
>
>
> Appendix Fig. 8 (b) shows comparisons on the Tanks\&Temples dataset. In this more complex real-world dataset, our ACRF outperforms all other algorithms. Although TensoRF still achieves a better rendering quality than DVGO (28.56 dB versus 28.31 dB in PSNR), the relatively large distortion introduced by Rho et al. places this TensoRF-based algorithm behind both ACRF and VQRF. Notably, our lightweight model, ACRF-F, achieves a similar compression result to Rho et al, while our ACRF-F requires only a few seconds for encoding, while Rho et al. demands 24 mins for training (see Table 1 in appendix of Rho et al.).
>
>
> **Q3. typo**
>
> We thank reviewer for the detailed comment. We have fixed the typo in our revised manuscript.

---

### Official Review · Reviewer_MUJE · 2023-11-04

**Soundness:** 3 good
**Presentation:** 2 fair
**Contribution:** 2 fair
**Rating:** 6
**Confidence:** 4

**Summary:**

This work addresses the problem of compressing explicit Neural Radiance Fields (NeRFs) for 3D data representation. The authors introduce a framework called Attributed Compression of Radiance Field (ACRF) to achieve this. ACRF prunes the neural 3D structure and encodes it as points with features using importance-guided encoding. It also employs an importance-based entropy model to optimize the encoding process. The authors present two models, ACRF and ACRF-F, balancing compression performance and encoding time. Experiments on synthetic and real-world datasets, including Synthetic-NeRF and Tanks&Temples, showcase the superior performance of their approach.

**Strengths:**

- The idea is simple yet effective.
- The experimental results are convincing and the ablation study shows the necessity of each component.

**Weaknesses:**

Major
- The proposed method is based on voxel-grid compression of NeRF with limited change so the novelty of it is limited.
- The paper writing can be further improved. For instance:
  - In section 3, the concrete definitions of r and N are not shown.
  - The motivation for adopting entropy minimization mentioned in section 4.3 is unclear.
Minor:
- I recommend the authors mention which section in the supplementary describes the details in the main paper.

**Questions:**

Major:
- Could you provide more elaboration about why RAHT with point importance introduces additional high-frequency noise to the original features and necessitates the transmission of importance values for decoding?
- Could the author provide more explanations for the motivation for adopting entropy minimization?
- How can the \labmda be tuned to control the model size?

Minor:
- Why do the authors only conduct the ablation study on the chair of the Synthetic-NeRF dataset? Could the authors provide experimental results of the ablation study on different datasets?

------------------------------------------------
### Post rebuttal:

Most of my concerns are addressed after reviewing the authors' responses and discussion between authors and reviewers. I am willing to raise my evaluation from 5 to 6.

---

> ### Author Response · Authors · 2023-11-22
> **Response to Reviewer MUJE**
>
> Thank you for your time and helpful feedback. We respond below to your questions and concerns. Please note that additional figures are included in the updated version of the appendix (Supplementary Material).
>
> **Q1. Novelty.**
>
> First and foremost, we would like to underscore the value of our work:
>
> 1. Our study presents a comprehensive analysis of explicit Neural Radiance Fields (NeRF) models, transforming the NeRF compression task into a paradigm of 3D data compression.
>
> 2. To the best of our knowledge, our ACRF is the first NeRF compression framework built upon 3D data compression techniques, including integration of pruning, position/feature coding and entropy coding.
>
> 3. The enhancement of our algorithm is realized through the incorporation of several modules, including view-dependent pruning, importance-guided feature encoding and entropy modeling.
>
> Thus, we believe that our ACRF constitutes a task-oriented algorithm with effective improvements.
>
>
>
> **Q2. The concrete definitions of r and N.**
>
> We thank reviewer for the detailed comments. We have updated a revision of the background section (Section 3), aligning it more closely with the original NeRF paper. Specifically, we update 'r' as the ray and 'N' as the number of sampled intervals along the ray.
>
>
> **Q3. The motivation of entropy minimization.**
>
> We rewrite the motivation of entropy minimization in section 4.3 as follows.
>
>
> According to information theory, the information entropy $H(\mathbf{e})$ of coefficients $\mathbf{e}$ indicates the average level of 'information' as $H(\mathbf{e}) = \mathbb{E}\left[-\log _2 p\left(\mathbf{e}\right)\right]$, where $p(\mathbf{e})$ is the probability distribution of coefficients $\mathbf{e}$. The objective of this section is to minimize the information entropy $H(\mathbf{e})$ inherent to the NeRF model through imposing constraints on the distribution $p(\mathbf{e})$, thereby reducing the model size of NeRF. Inspired by deep image compression [1], we learn an estimated distribution $q(\mathbf{e})$ to constrain $p(\mathbf{e})$. It is also worth noting that there exists a strong interrelation between the distribution of point features and their corresponding importance $I_l$. Thus, we further introduce $I_l$ as a prior information component.
>
>
>
> [1] Variational image compression with a scale hyperprior.
>
>
> **Q4. Appendix section number for the main paper.**
>
> We thank reviewer for the detailed comment. We have added appendix section number for the main paper.
>
>
> **Q5. Elaboration about straightforward implementation of Importance-Guided Feature Encoding.**
>
> We provide more elaborations about straightforward implementation of importance-guided feature encoding in appendix Sec. H as follows.
>
> Given point positions $\mathbf{p}$ and features $\mathbf{f}$, we transform features into coefficients through RAHT transform as $\mathbf{e}=\text{RAHT}(\mathbf{p}, \mathbf{f})$. Details about RAHT are provided in appendix Sec. D.
>
> Incorporating importance $I_l$ as prior information, our objective is to leverage $I_l$ to effectively preserve point features with heightened significance. In the initial implementation described in Sec. 4.2, we obtain coefficients as:
> $$\mathbf{e}=\text{RAHT}(\mathbf{p}, \mathbf{f}_I), \quad \mathbf{f}_I=\mathbf{f} \cdot I_l.$$Throughout the decoding process, the reconstructed features $\hat{\mathbf{f}}$ can be recovered from the inverse RAHT transform while considering importance:
> $$\hat{\mathbf{f}}=\frac{\hat{\mathbf{f}}_I}{I_l}, \quad \hat{\mathbf{f}}_I=\text{iRAHT}(\mathbf{p}, \mathbf{e}).$$Note that, for the recovery of point features, the transmission of importance values is requisite for decoding.
>
> We further provide an examination of the high-frequency noise issue. As shown in appendix Fig. 6 (a) and (b), we visualize the GT image, view-independent and view-independent importance, and the top 1\% point features sorted by their corresponding importance. Importance is visualized as a heatmap, where red signifies higher values and blue lower ones, and features are projected into color space through t-SNE.
>
>
> In appendix Fig. 6 (a), the point feature representation of the black stick of the lego in the GT image appears as green in the projected feature image. However, in the importance figure, it is composed of red, yellow, and blue elements. A similar observation is also shown in appendix Fig. 6 (b). This illustrates that, unlike point features, point importance lacks a robust correlation with the scene color. Consequently, the direct combination of features with importance may introduce additional noise to the original features.

---

> > ### Author Response · Authors · 2023-11-22
> > **Response to Reviewer MUJE (2)**
> >
> > **Q6. The motivation of entropy minimization.**
> >
> > Similar to Q3.
> >
> >
> > **Q7. How can the labmda be tuned to control the model size?**
> >
> >
> > In a manner akin to some practical compression tools, such as ffmpeg, we manually tune the hyperparameter lambda to strike a balance between the model size and rendering quality. This tuning process is similar to factors like CRF and QP in ffmpeg. In our experiments, we empirically set lambda along with other hyperparameters, as detailed in appendix Sec. F.
> >
> >
> >
> > **Q8. Ablation studies on Synthetic-NeRF.**
> >
> > We conduct ablation studies on chair only for simplicity. We provide ablation studies on the whole Synthetic-NeRF dataset in appendix Sec. I as follows.
> >
> >
> > Ablation studies on the whole Synthetic-NeRF dataset are provided in appendix Fig. 7, and these findings are consistent with the results obtained for the chair.

---

### Meta-Review · Area_Chair_dfrB · 2023-12-10

**Metareview:**

The authors reformulate the task of explicit NeRF compression as 3D data compression and introduce a NeRF compression framework, Attributed Compression of Radiance Field (ACRF). ACRF prunes the neural 3D structure and converts it to points with features further encoded using importance-guided feature encoding. ACRF also employs an importance-based entropy model for the encoding process. The experiments show the advantages of the proposed framework.

The interesting idea and the effectiveness of the proposed method are emphasized by most of the reviewers, while limited novelty at the level of basic employed techniques can be seen as a major weakness.

The authors provided responses to all the reviewers and the reviewers are generally positive in recommending acceptance of the paper (6,6,8,8).

The meta-reviewer after carefully reading the reviews, the discussions, and the paper, agrees with the reviewers and recommends acceptance.

**Justification For Why Not Higher Score:**

While the idea and the effectiveness of the method are strengths, the advances at the machine learning or theoretical level are limited and the general interest for the addressed topic is limited as well.

**Justification For Why Not Lower Score:**

Four reviewers and the meta-reviewer agree that the paper has merits, and the contributions are sufficient and of interest. There is no significant flaw to impede publication.

---

### Decision · Program_Chairs · 2024-01-16

Accept (poster)